# Biallelic JAK1 mutations in immunodeficient patient with mycobacterial infection

Davide Eletto[1,*], Siobhan O. Burns[2,3,*], Ivan Angulo[1], Vincent Plagnol[4], Kimberly C. Gilmour[5], Frances Henriquez[5], James Curtis[1], Miguel Gaspar[1], Karolin Nowak[6], Vanessa Daza-Cajigal[2], Dinakantha Kumararatne[7], Rainer Doffinger[7], Adrian J. Thrasher[5,6,**] & Sergey Nejentsev[1,**]

Mutations in genes encoding components of the immune system cause primary immuno-deficiencies. Here, we study a patient with recurrent atypical mycobacterial infection and early-onset metastatic bladder carcinoma. Exome sequencing identified two homozygous missense germline mutations, P733L and P832S, in the JAK1 protein that mediates signalling from multiple cytokine receptors. Cells from this patient exhibit reduced JAK1 and STAT phosphorylation following cytokine stimulations, reduced induction of expression of inter-feron-regulated genes and dysregulated cytokine production; which are indicative of signalling defects in multiple immune response pathways including Interferon-γ production. Recon-stitution experiments in the JAK1-deficient cells demonstrate that the impaired JAK1 function is mainly attributable to the effect of the P733L mutation. Further analyses of the mutant protein reveal a phosphorylation-independent role of JAK1 in signal transduction. These findings clarify JAK1 signalling mechanisms and demonstrate a critical function of JAK1 in protection against mycobacterial infection and possibly the immunological surveillance of cancer.

[1] Department of Medicine, University of Cambridge, Cambridge CB2 0QQ, UK. [2] University College London Institute of Immunity and Transplantation, London NW3 2PF, UK. [3] Department of Immunology, Royal Free London NHS Foundation Trust, London NW3 2PF, UK. [4] University College London Genetics Institute, University College London, London WC1E 6BT, UK. [5] Great Ormond Street Hospital for Children NHS Foundation Trust, London WC1N 3JH, UK. [6] University College London Institute of Child Health, London WC1N 1EH, UK. [7] Department of Clinical Biochemistry and Immunology, Addenbrooke's Hospital, Cambridge CB2 2QQ, UK. * These authors contributed equally to this work. ** These authors jointly supervised the work. Correspondence and requests for materials should be addressed to S.N. (email: sn262@cam.ac.uk).

Primary immunodeficiencies (PIDs) are genetic disorders that cause immune dysfunction and predisposition to infection. Selective susceptibility to weakly virulent mycobacteria, such as M. bovis Bacillus Calmette-Guerin vaccine or environmental mycobacteria species, is a genetically heterogeneous group of rare PIDs so far associated with mutations in nine genes (IFNGR1, IFNGR2, IL12B, IL12RB1, STAT1, ISG15, IRF8, IKBKG and CYBB)[1]. These mutations impair the production of or the response to a cytokine Interferon-γ (IFN-γ), either directly or indirectly, indicating that the IFN-γ pathway is critical for the confinement of mycobacterial infection[2]. Nevertheless, genetic aetiology in approximately half of patients with Mendelian susceptibility to mycobacterial diseases remains unknown[3].

IFN-γ is a type II interferon that binds to the IFN-γ receptor, a heterodimer encoded by genes IFNGR1 and IFNGR2. Stimulation of the IFN-γ receptor results in the downstream activation of two Janus kinases: JAK1 and JAK2. Upon activation, JAKs trans-phosphorylate each other at tyrosines within the kinase domain and phosphorylate the cytoplasmic tail of the receptor[4]. This allows recruitment of the Signal Transducer and Activator of Transcription 1 (STAT1) protein, which in turn is phosphorylated, forms homodimers, relocates to the nucleus, binds the Gamma Activated Sequences in the genome and drives the expression of genes implicated in cellular immunity, including antigen processing and presentation and activation of microbicidal effector functions. Intracellular signalling of type I interferons, for example, IFN-α and IFN-β, is mediated by the Interferon-α receptor encoded by IFNAR1 and IFNAR2. The receptor interacts with Janus kinases JAK1 and TYK2, leading to phosphorylation of STAT1 and STAT2 proteins, which then form a heterodimer that translocates to the nucleus, forms a complex with Interferon Regulatory Factor 9 and induces the expression of the interferon-stimulated genes[5]. Multiple other cytokine receptors also signal through combinations of four JAKs and seven STAT proteins, for example, JAK1 is also used in signalling by IL-2, IL-4, IL-7, IL-9, IL-15, IL-21, IL-27, IL-6 family cytokines and IL-10 family cytokines[4]. To date, germline mutations in two out of the four known Janus kinases, JAK3 and TYK2, have been found in PID patients[6–9]. Somatic mutations in JAK2 have also been shown to cause clonal myeloproliferative disorders, for example, polycythemia vera and idiopathic erythrocytosis[10,11], whereas somatic JAK1 mutations have been associated with gynaecologic cancers[12].

Here, we report the identification of germline JAK1 mutations that result in a functional JAK1 deficiency associated with susceptibility to atypical mycobacterial infection and early-onset bladder carcinoma. Furthermore, detailed analyses of the mutant protein reveal phosphorylation-independent mechanism of JAK1 in signal transduction.

## Results

**Immunodeficiency with susceptibility to mycobacteria.** We studied a 22-year-old male of Pakistani descent, the last of four children born to a consanguineous marriage of first cousins (Fig. 1a). The patient presented to paediatric immunology at the age of 3 years with a history of global developmental delay and recurrent ear and chest infections that started during the first year of life and required multiple hospital admissions. The patient had received childhood vaccines—including Bacillus Calmette-Guerin vaccine at birth—and had normal-course chicken pox at age 3 with one subsequent episode of shingles. During examination, a skeletal survey demonstrated lytic and sclerotic lesions affecting long bones, vertebrae and facial bones. The patient also developed cervical

lymphadenopathy. Bone biopsy was unremarkable, lymph node biopsy reactive and no pathogen was cultured from either tissue. Considering that these features were associated with failure to thrive, raised erythrocyte sedimentation rate (ESR) (70–90 mm per hr), elevated polyclonal IgG (25–30 g l$^{-1}$), platelets (600–700 × 10$^9$ l$^{-1}$) and white cell count (20–25 × 10$^9$ l$^{-1}$), he was further investigated for infection. Mycobacterial skin tests for Mycobacterium avium and Mycobacterium intracellulare were negative, but Mycobacterium tuberculosis (Mtb), Mycobacterium malmoense and Mycobacterium scrofulaceum skin tests were all strongly positive. As his relatively indolent clinical course was not typical for tuberculosis and Mtb had not been cultured from bone or lymph node, a clinical diagnosis of systemic atypical mycobacterial infection was made. He received anti-mycobacterial treatment (Isoniazid, Ethambutol and Ciprofloxacin, as other agents were not tolerated) and his condition improved over 12 months with catch-up growth (from 3rd to 25th centile), weight gain (25th to 50th centile) and improvement in ESR, IgG, platelets and white cell count. His bone X-rays also showed improvement with residual vertebral collapse, supporting a diagnosis of resolving multifocal osteomyelitis caused by mycobacterial infection. The immunology investigations demonstrated normal numbers of T and B cells, reduced populations of naive CD4+ and CD8+ T cells (Table 1) with normal proliferation after phytohemagglutinin (PHA) stimulation, and mildly reduced responses to Candida and purified protein derivative antigens. Total IgG and IgA levels were increased, whereas IgM level was normal, as were specific antibody responses after tetanus, Hib and pneumococcal vaccinations. The karyotype, metabolic screen and chromosomal radio-sensitivity assays were normal.

The patient remained relatively well until the age of 16, with mild developmental delay and short stature. His IgG levels remained high and over time IgM levels fell below the normal range (Table 1) with persistent mild T lymphopenia and impaired responses to PHA stimulation. Normal CDR3 spectratyping results in CD4+ and CD8+ T cells were found, with all TCR Vβ families represented with a Gaussian distribution. T-cell receptor excision circles levels in CD4+ and CD8+ T cells were normal.

At the age of 16 years the patient presented with unexplained cardiomyopathy and a raised ESR (20–40 mm h$^{-1}$) and was found to have a mediastinal mass on computed tomography imaging. Biopsies showed pleural and mediastinal fibrosis with patches of macrophage infiltration in lung tissue. No granulomas were seen and Quantiferon TB Gold test was negative. Mycobacterium gordonae was isolated from a single sputum sample, but its relevance remained unclear. In view of his previous history he received empiric treatment for atypical mycobacteria (Rifampicin and Ethambutol, Clarithromycin and Ciprofloxacin) with improvement of ESR, IgG level and the mass but permanent presumed fibrotic occlusion of the right pulmonary vein. He remained on long-term prophylaxis with Clarithromycin and Ciprofloxacin and had no recurrence or further mycobacterial infections. He had a number of skin infections, including planar warts restricted to the forehead, presumed fungal infections of his nails and severe Norwegian scabies.

At the age of 21 years the patient developed significant anaemia. He had a history of intermittent red blood per rectum and no recent history of haematuria. Colonoscopy revealed a large sessile polyp in his rectum, which was histologically benign without dysplasia. Thickening of the bladder wall was noted on magnetic resonance imaging and an extensive fungating tumour was observed on cystoscopy. Biopsies of the tumour and a supraclavicular lymph node confirmed

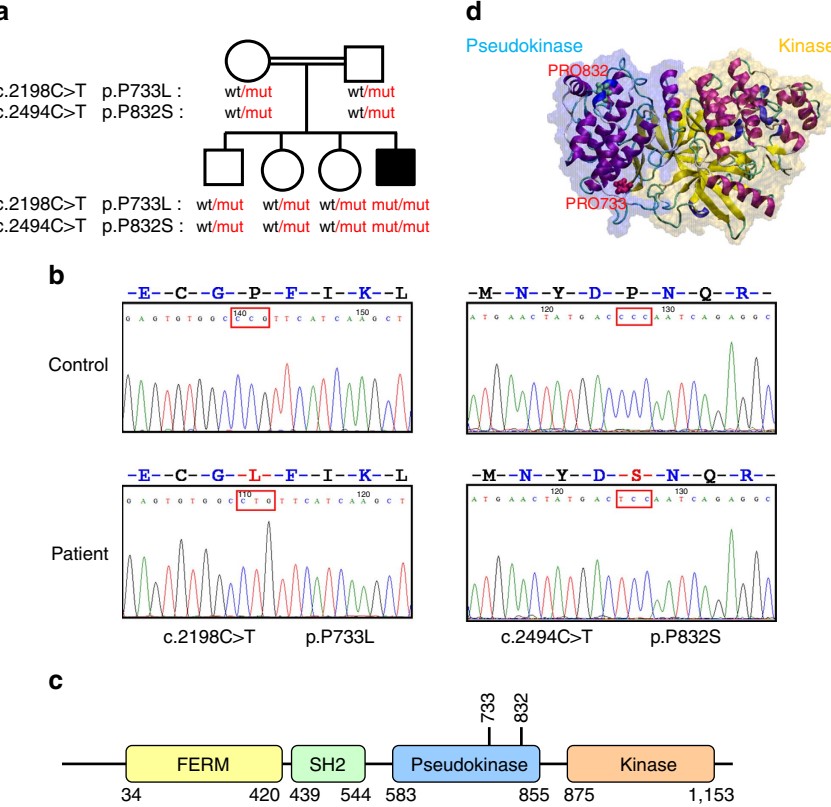

**Figure 1 | Two JAK1 mutations found in the patient.** (**a**) Patient's family tree. (**b**) Sequence chromatograms showing two mutations in the JAK1 gene. (**c**) Domain structure of the JAK1 protein. (**d**) JAK1 pseudokinase (JH2) and kinase (JH1) domains modelled using the published structure of TYK2 pseudokinase and kinase domains (PDB 4OLI). Pro733 and Pro832 are shown in red and green, respectively.

high-grade metastatic transitional cell carcinoma. The patient received treatment with chemotherapy including cisplatin and gemcitabine, but died aged 23 years.

**Exome sequencing identifies homozygous *JAK1* mutations.** As the patient was born to a consanguineous family, we hypothesized that his disease was caused by a recessive Mendelian mutation and investigated this possibility by whole-exome sequencing. The exome data contained 18,029 single nucleotide variants and small insertions/deletions, including 230 very rare ones that were not observed in the 6,500 NHLBI Exomes[13], 1,000 Genomes database (April 2012 data release)[14] and 2,500 exomes analysed internally using the same bioinformatics pipeline. Five of these rare variants were homozygous (Supplementary Table 1). Of these five, three were predicted to be benign, whereas two missense mutations were predicted to be probably damaging; both were located in the *JAK1* gene, leading to amino-acid changes from proline to leucine (p.P733L) and from proline to serine (p.P832S) (Fig. 1b). We found no mutations in genes previously associated with Mendelian susceptibility to mycobacterial disease, or other genes involved in the JAK–STAT signalling pathways. We confirmed both *JAK1* mutations by Sanger sequencing (Fig. 1b) and found that unaffected parents and all three siblings were heterozygous carriers of both mutations (Fig. 1a). We then designed genotyping assays for both *JAK1* mutations, screened 1,050 healthy subjects representing 51 populations from around the world[15] and found no healthy carriers. The ExAC database that contains exome data from > 60,000 subjects had no instance of the p.P733L mutation, whereas

p.P832S was detected in four heterozygous individuals (frequency = 0.000033).

Similarly to other Janus kinases, JAK1 has FERM and SH2 domains that are responsible for interaction with the cytokine receptor, the pseudokinase (JH2) domain that regulates kinase activity, and the kinase (JH1) domain[16,17]. The proline residues at JAK1 positions 733 and 832 are located in the pseudokinase domain (Fig. 1c). They are conserved within the human Janus kinase family and in JAK1 across species (Supplementary Fig. 1). We visualized both mutations by modelling JAK1 pseudokinase and kinase domains on the published TYK2 structure[18]. Although P832S was located far from the kinase domain, P733L mapped in the β7–β8 loop close to the inter-domain interface and may affect interaction between the domains (Fig. 1d). Taken together, these results suggest that the identified JAK1 genetic variants, P733L in particular, could be recessive pathogenic mutations rather than rare neutral polymorphisms.

**Multiple affected JAK1-mediated pathways in immune cells.** To test the hypothesis that JAK1 P733L and P832S mutations are pathogenic, we studied STAT phosphorylation in the patient's lymphocytes (Fig. 2a). STAT1 phosphorylation was significantly reduced after IFN-α, IFN-γ and IL-27 stimulations. STAT3 phosphorylation was reduced after IL-10—but not IL-6—stimulation. Phosphorylation of STAT4 was reduced after IFN-α stimulation, as were phosphorylation of STAT5 after IL-2 stimulation and of STAT6 after IL-4 stimulation (Fig. 2a). Therefore, multiple signalling pathways mediated by JAK1 are affected in the patient's immune cells, suggesting a functional JAK1 deficiency.

**Table 1 | Immunological investigations.**

| Cell type/Ig class | Patient (3 years 8 months) | Age-matched control range | Patient (10 years) | Age-matched control range | Patient (19 years) | Age-matched control range |
|---|---|---|---|---|---|---|
| White cell count | *$26.8 \times 10^9 \, l^{-1}$ | $5.0$-$15.0 \times 10^9 \, l^{-1}$ | $4.95 \times 10^9 \, l^{-1}$ | $4.5$-$13.5 \times 10^9 \, l^{-1}$ | $7.8 \times 10^9 \, l^{-1}$ | $4$-$11.9 \times 10^9 \, l^{-1}$ |
| Neutrophil count | *$17.78 \times 10^9 \, l^{-1}$ | $1.0$-$8.5 \times 10^9 \, l^{-1}$ | $2.47 \times 10^9 \, l^{-1}$ | $1.8$-$8.0 \times 10^9 \, l^{-1}$ | $4.9 \times 10^9 \, l^{-1}$ | $2$-$7.5 \times 10^9 \, l^{-1}$ |
| Lymphocyte count | $6.08 \times 10^9 \, l^{-1}$ | $3.0$-$13.5 \times 10^9 \, l^{-1}$ | $1.62 \times 10^9 \, l^{-1}$ | $1.1$-$5.9 \times 10^9 \, l^{-1}$ | $1.55 \times 10^9 \, l^{-1}$ | $1$-$2.8 \times 10^9 \, l^{-1}$ |
| CD3+ T cells | 49%, $3.0 \times 10^9 \, l^{-1}$ | 39-73%, $1.8$-$8.0 \times 10^9 \, l^{-1}$ | *45%, $0.73 \times 10^9 \, l^{-1}$ | 55-78%, $0.7$-$4.2 \times 10^9 \, l^{-1}$ | *29%, *$0.45 \times 10^9 \, l^{-1}$ | 55-83%, $0.7$-$2.1 \times 10^9 \, l^{-1}$ |
| CD19+ B cells | *45%, $2.7 \times 10^9 \, l^{-1}$ | 17-41%, $0.6$-$3.1 \times 10^9 \, l^{-1}$ | *33%, $0.53 \times 10^9 \, l^{-1}$ | 10-31%, $0.2$-$1.6 \times 10^9 \, l^{-1}$ | *40%, $0.62 \times 10^9 \, l^{-1}$ | 6-19%, $0.1$-$0.5 \times 10^9 \, l^{-1}$ |
| CD16+CD56+ NK cells | 3%, $0.2 \times 10^9 \, l^{-1}$ | 3-16%, $0.1$-$1.4 \times 10^9 \, l^{-1}$ | 20%, $0.32 \times 10^9 \, l^{-1}$ | 4-26%, $0.09$-$0.9 \times 10^9 \, l^{-1}$ | 28%, $0.43 \times 10^9 \, l^{-1}$ | 7-13%, $0.09$-$0.6 \times 10^9 \, l^{-1}$ |
| CD3+CD4+ T cells | 28%, $1.7 \times 10^9 \, l^{-1}$ | 25-50%, $0.9$-$5.5 \times 10^9 \, l^{-1}$ | 23%, $0.37 \times 10^9 \, l^{-1}$ | 27-53%, $0.3$-$2.0 \times 10^9 \, l^{-1}$ | *16%, *$0.25 \times 10^9 \, l^{-1}$ | 28-57%, $0.3$-$1.4 \times 10^9 \, l^{-1}$ |
| CD3+CD8+ T cells | 23%, $1.4 \times 10^9 \, l^{-1}$ | 11-32%, $0.4$-$2.3 \times 10^9 \, l^{-1}$ | 16%, $0.26 \times 10^9 \, l^{-1}$ | 19-34%, $0.3$-$1.8 \times 10^9 \, l^{-1}$ | 12%, *$0.19 \times 10^9 \, l^{-1}$ | 10-39%, $0.2$-$0.9 \times 10^9 \, l^{-1}$ |
| CD4+CD45RA+ T cells | *12% | 62-90 % | | | 43%[†] | 31-65% |
| CD8+CD45RA+ T cells | *9% | 46-85 % | | | *24%[†] | 42-73% |
| γδT-cell | 2% | <10% | | | | |
| IgG | *$38.8 \, g \, l^{-1}$ | $3.1$-$13.8 \, g \, l^{-1}$ | *17.8 | 5.4-16.1 | $11.9 \, g \, l^{-1}$ | 6.0-16.0 |
| IgA | *$1.8 \, g \, l^{-1}$ | $0.3$-$1.2 \, g \, l^{-1}$ | 1.00 | 0.7-2.5 | $0.9 \, g \, l^{-1}$ | 0.8-2.8 |
| IgM | $1.5 \, g \, l^{-1}$ | $0.5$-$2.2 \, g \, l^{-1}$ | 0.61 | 0.5-1.8 | *$0.45 \, g \, l^{-1}$ | 0.5-1.9 |
| IgG1 | | | *12.9 | 3.6-7.3 | | |
| IgG2 | | | *0.82 | 1.4-4.5 | | |
| IgG3 | | | 0.77 | 0.3-1.1 | | |
| PHA stimulation[‡] | 172 | ≥70 | *12.6 | ≥70 | *3.32 | ≥14.4 |
| Candida stimulation[‡] | *17.6 | ≥26.8 | | | | |
| PPD stimulation[‡] | *26.2 | ≥41.5 | | | | |
| CD3 stimulation[‡] | | | | | *3.2 | ≥7.6 |

*Denotes an abnormal result.
[†]Results are shown for the CD45RA+CD27+ cells.
[‡]T-cell stimulation index calculated as maximum stimulation value/background value.

We then stimulated whole blood from the patient and measured production of cytokines IFN-γ, IL-12, IL-10, TNF-α and IL-6. In assays normalized for T-cell counts we found consistently low IFN-γ production in response to PHA stimulation and after co-stimulations with PHA/IL-12 or PHA/IFN-α in comparison with healthy controls (Fig. 2b). However, upregulation of IFN-γ production after co-stimulations with lipopolysaccharide/interleukin-12 (LPS/IL-12) or LPS/IFN-α was normal. Also, the patient had low IL-10 production after PHA stimulation and co-stimulation with PHA/IFN-α (Fig. 2c). Production of IL-6 and TNF-α was increased after LPS stimulation (Fig. 2d,e), whereas production of IL-12 was normal (Fig. 2f). These results indicate that functional JAK1 deficiency is characterized by a broad immune dysregulation.

**Impaired phosphorylation of JAK1, partner JAKs and STATs.** To understand how the P733L and P832S mutations affect JAK1, we next looked at IFN-α and IFN-γ signalling in patient's primary fibroblasts. We first tested the hypothesis that mutations impact on the stability of the JAK1 protein, affecting, in turn, its intracellular levels. We measured JAK1 in fibroblasts derived from the patient or from two healthy subjects (Supplementary Fig. 2a), and in HEK-239T cells transiently expressing the wild type JAK1$^{WT}$, the patient-derived double-mutant JAK1$^{P733L/P832S}$ or the kinase-dead mutant JAK1$^{K908E}$ (Supplementary Fig. 2b). The patient-derived variant of JAK1 was expressed at a slightly lower level than the wild-type JAK1.

The JAK pseudokinase domain regulates activity of the kinase domain. Hence, both engineered and naturally occurring mutations in the pseudokinase domain of various JAKs, including known somatic mutations in JAK1, can affect the kinase activity[19–22]. To investigate functional effects of the P733L and

P832S mutations, we measured JAK1-mediated activation of STAT1 and STAT2 proteins in response to stimulation with cytokines IFN-α and IFN-γ. The level of phosphorylation of JAK1 upon exposure to IFN-α was profoundly reduced in patient-derived fibroblasts as compared with control cells (Fig. 3a). Phosphorylation of STAT1 and STAT2 was also impaired (Fig. 3a). A reduced STAT1 phosphorylation was also observed upon treatment with IFN-γ (Fig. 3b). The induction of expression of interferon-regulated genes was also lower in the patient fibroblasts compared with control fibroblasts (Fig. 4 and Supplementary Fig. 3). These data demonstrate that not only patient's peripheral blood mononuclear cells, but also primary fibroblasts show impaired JAK1 functions, leading to reduced downstream STAT signalling; such fibroblasts therefore provide a suitable model for the analysis of JAK1 functions.

The JAK1 pseudokinase domain keeps the basal activity of the kinase in check and mediates the cytokine-inducible activation of signalling[20]. The mutations found in the patient could either affect the level of JAK1 phosphorylation or its onset or decay, causing delayed or shortened cellular responses. We studied responses to IFN-α and IFN-γ in a time-course experiment and observed reduced levels of phosphorylated JAK1, as well as STAT1, with no effect on the duration or the steepness of the activation/inactivation phases (Fig. 5a,b). These results suggest that the phospho-transfer function of the mutant JAK1 was impaired, whereas the basal kinetics of phosphorylation and dephosphorylation were normal.

We then studied if the mutant JAK1 affected phosphorylation of the partner Janus kinases TYK2 and JAK2 after stimulation with IFN-α and IFN-γ, respectively. We found that in patient's fibroblasts the amount of phosphorylated TYK2 was strongly reduced, whereas the amount of phosphorylated JAK2 was only slightly diminished (Fig. 6).

**P733L has stronger effect on signalling than P832S.** To study effects of each of the two patient's mutations separately, we cloned the wild-type JAK1 (JAK1$^{WT}$) and introduced P733L, P832S or P733L/P832S mutations by site-directed mutagenesis.

We then expressed these constructs in the human fibrosarcoma U4A cells that lack endogenous JAK1 (ref. 23). The robustness of this model relies on the fact that STAT1 and STAT2 phosphorylation after stimulation with interferons is totally

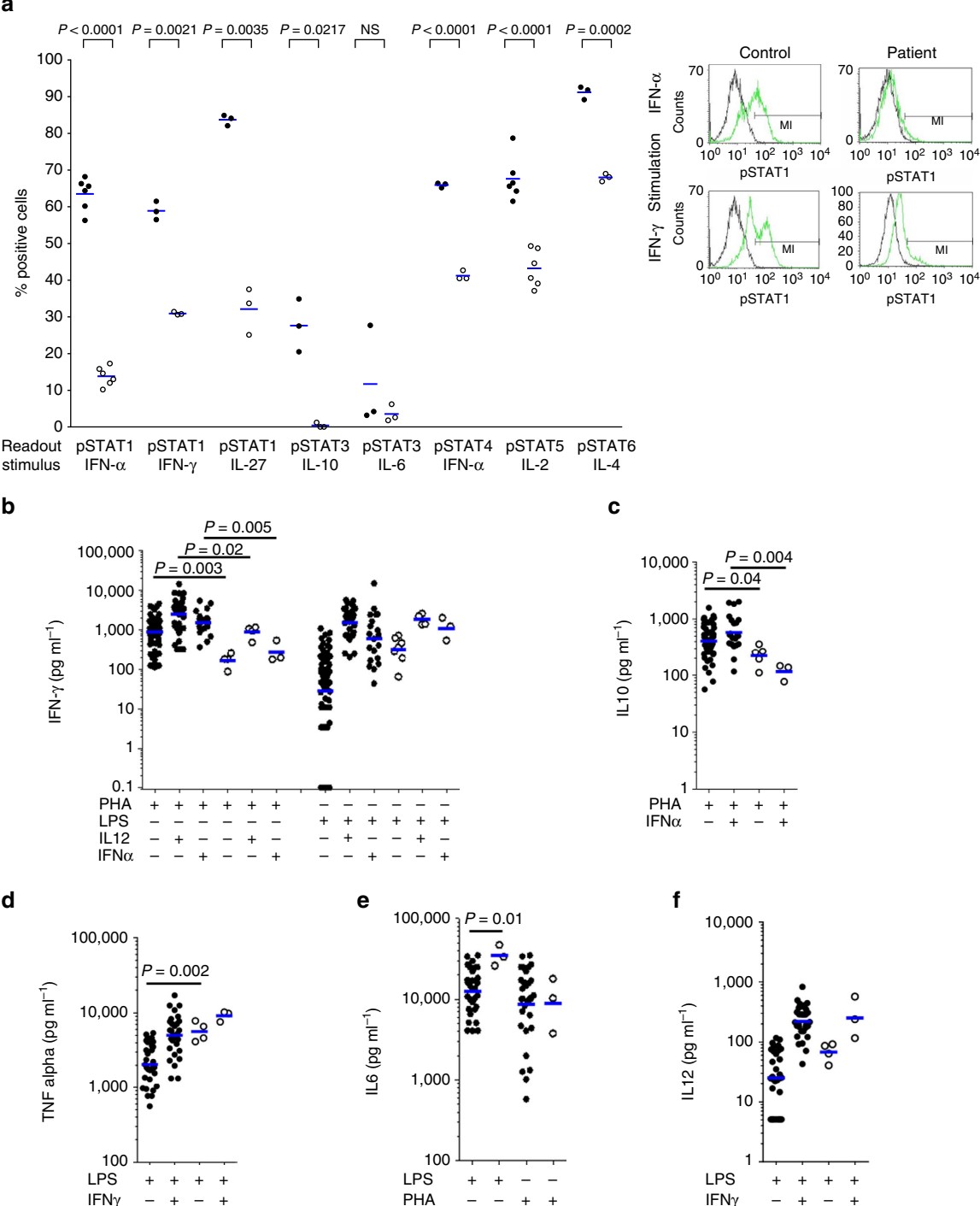

**Figure 2 | Impaired STAT phosphorylation and cytokine responses in the patient's blood cells.** (**a**) Left panel. Numbers of cells positive for the presence of phosphorylated STAT proteins were measured by FACS after 10 min stimulation of whole blood of the patient (age 20 years) (open circles) and compared with a healthy travel control tested under the same conditions (black dots). The assay was repeated either three times or six times (thrice on two occasions, in which case two different travel controls were studied). Blue lines show geometric means. Unpaired two-tailed Student $t$-test with Welch's correction. Right panel. FACS gating for pSTAT1 after IFN-α and IFN-γ stimulation is shown. (**b–f**) Cytokine responses measured after stimulation in whole blood of the patient in independent assays (open circles) and compared with healthy controls tested under the same conditions (black dots). Numbers of controls in different assays were: (**b**) LPS only $n = 65$, LPS + IL-12 $n = 40$, LPS + IFN-α $n = 20$, PHA only $n = 60$, PHA + IL-12 $n = 45$, PHA + IFN-α $n = 20$; (**c**) PHA only $n = 50$, PHA + IFN-α $n = 20$; (**d–f**) $n = 30$. Blue lines show geometric means. Two-tailed Mann–Whitney test.

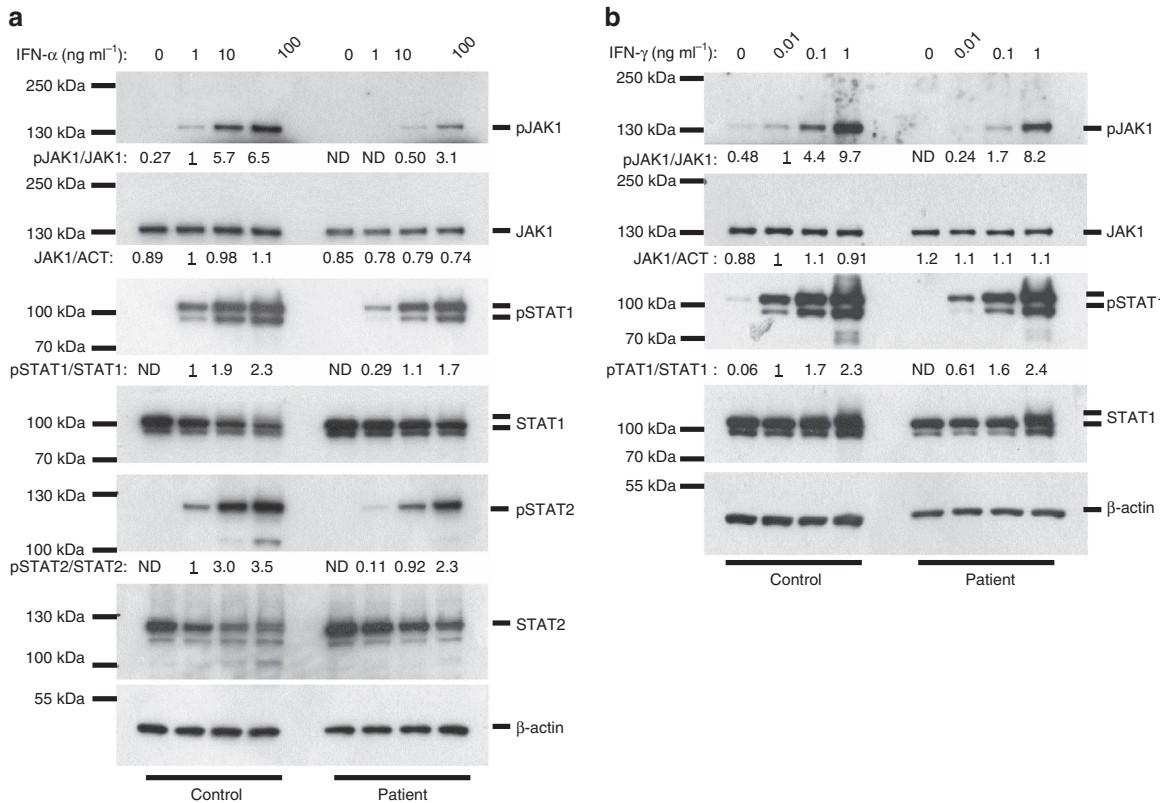

**Figure 3 | Impaired JAK1-mediated signalling in the patient's fibroblasts.** (**a**,**b**) Primary fibroblasts from the patient or a healthy control were treated with the indicated concentrations of IFN-α (**a**) or IFN-γ (**b**) for 15 min and protein extracts were subjected to immunoblotting. Representative of three independent experiments. Fold change of band densitometry is indicated (numbers below bands and bar graphs in Supplementary Fig. 6).

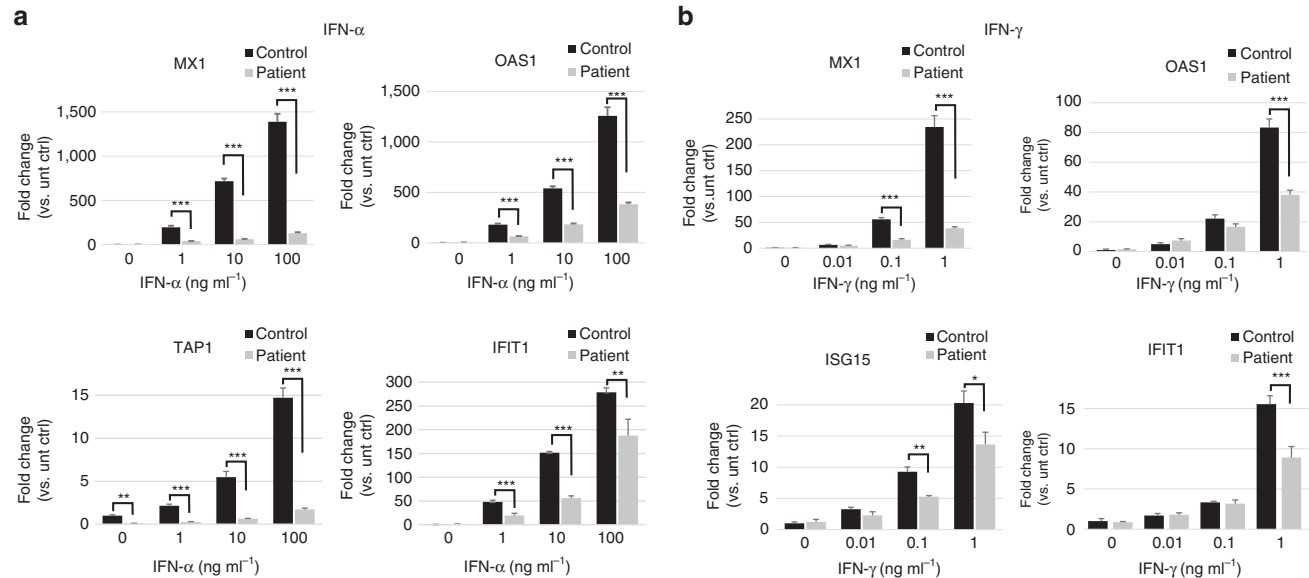

**Figure 4 | Patient's fibroblasts show reduced induction of gene expression after IFN-α and IFN-γ stimulation.** Cells were stimulated with IFN-α for 15 h (**a**) or IFN-γ for 8 h (**b**). qPCR was done in triplicate. mRNA fold change is shown relative to the untreated control fibroblasts. *$P < 0.05$, **$P < 0.005$, ***$P < 0.0005$. Graphs show mean values ± s.d.

dependent on JAK1 and in U4A cells the signalling pathway is rescued only upon re-expression of JAK1 (Fig. 7a,b)[23,24]. We found that phosphorylation of JAK1[P832S] was similar to that of JAK1[WT], but phosphorylation of JAK1[P733L/P832S] was completely abolished and that of JAK1[P733L] was either abolished (Fig. 7a) or very strongly reduced (Fig. 7b), suggesting that the amino-acid change P733L in the pseudokinase domain

impairs JAK1 function leading to the reduced phosphorylation of tyrosines Y1034 and Y1035 in the activation loop of the kinase domain (Fig. 7a,b). Nevertheless, all four JAK1 variants were able to mediate STAT1 and STAT2 phosphorylation (Fig. 7a,b).

We then transduced primary patient fibroblasts with lentiviral vectors expressing the wild-type and mutant JAK1 proteins (Fig. 7c). Forced expression of JAK1[WT] significantly increased

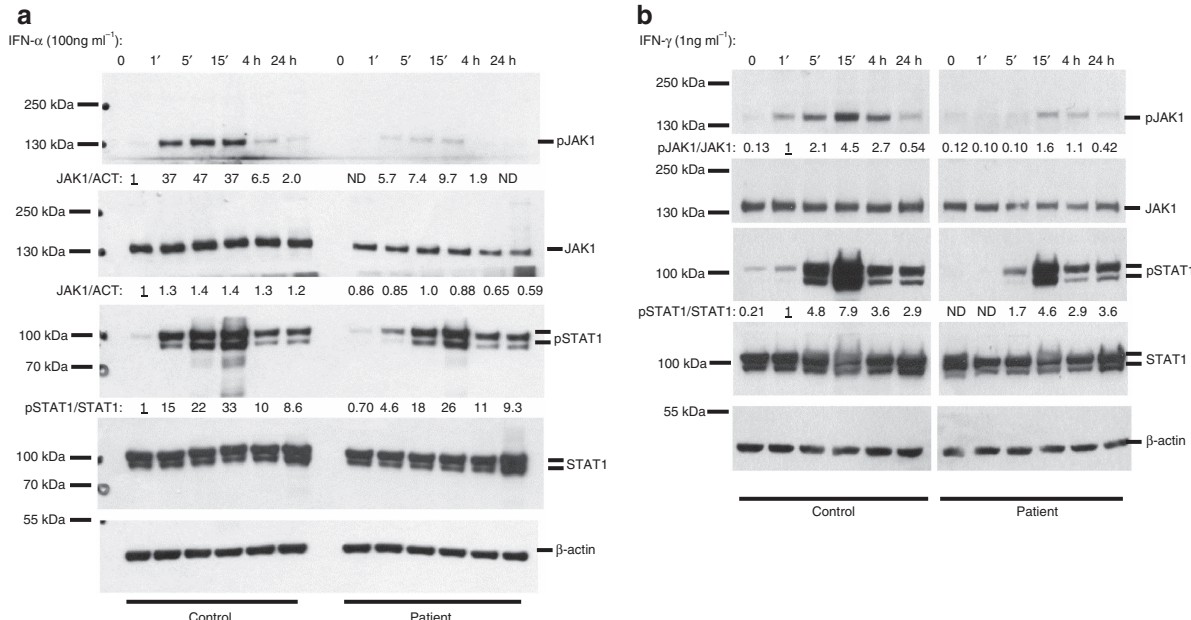

**Figure 5 | Reduced JAK1 and STAT1 phosphorylation in patient's fibroblasts at different time points after IFN-α and IFN-γ stimulation. (a,b)** Primary fibroblasts from the patient or a healthy control were treated with the IFN-α (**a**) or IFN-γ (**b**) for indicated times and protein extracts were subjected to immunoblotting. Representative of two independent experiments. Fold change of band densitometry is indicated (number below bands and bar graphs in Supplementary Fig. 7).

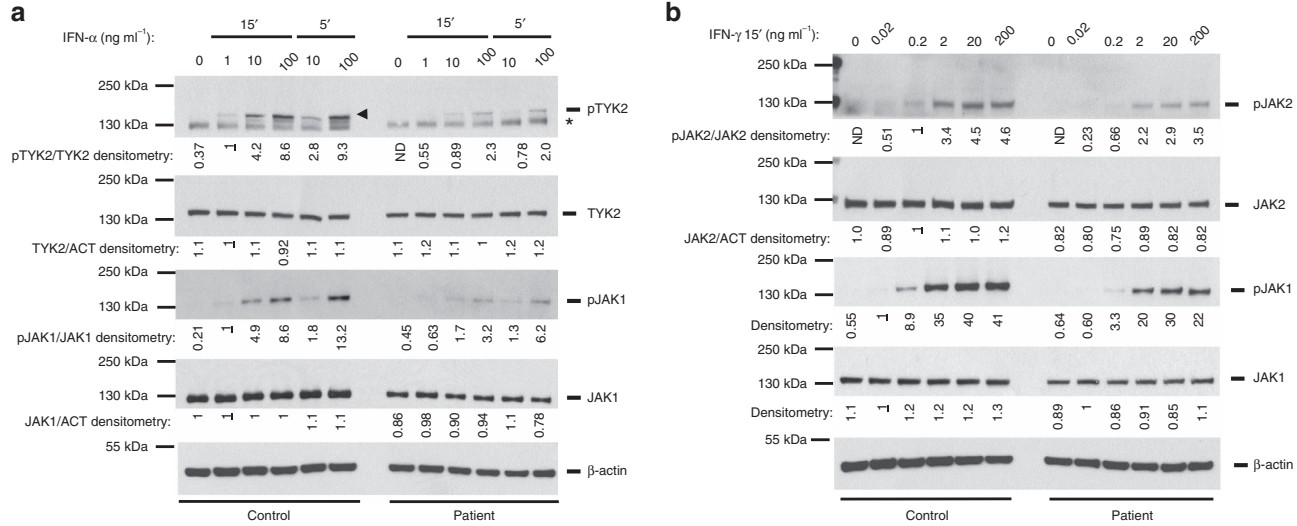

**Figure 6 | Reduced amounts of phosphorylated TYK2 and JAK2 in the patient's fibroblasts. (a,b)** Primary fibroblasts from the patient or a healthy control were treated with the indicated concentrations of IFN-α (**a**) or IFN-γ (**b**) and protein extracts were subjected to immunoblotting. *shows non-specific protein species detected by the antibody. Arrowhead shows specific bands. Representative of three independent experiments (IFN-α 15′ and IFN-γ) or one experiment (IFN-α 5′). Fold change of band densitometry is indicated (numbers below bands and bar graphs in Supplementary Fig. 8).

STAT1 phosphorylation after stimulation with IFN-α and IFN-γ. Forced expression of JAK1[P832S] rescued STAT1 phosphorylation to a similar extent as JAK1[WT], whereas expression of the JAK1[P733L] or JAK1[P733L/P832S] proteins led to significantly reduced STAT1 phosphorylation. Taken together, these results indicate that the double-mutant JAK1[P733L/P832S] protein is functionally deficient and that its defect is mostly caused by the P733L mutation, whereas the contribution of P832S is less pronounced.

**JAK1 can mediate signalling independently of its phosphorylation.** It is known that overexpression of JAKs can lead to self-

phosphorylation and activation[25]. This explains the observation that forced expression of JAK1[WT] or JAK1[P832S] in U4A cells caused spontaneous self-phosphorylation of JAK1, as well as phosphorylation of STAT1 and STAT2, even in the absence of interferon stimulation (Fig. 7a,b). Unexpectedly we observed that following interferon stimulation, JAK1[P733L/P832S] was able to induce STAT1 and STAT2 phosphorylation even in the absence of JAK1 phosphorylation (Fig. 7a,b). Similarly, JAK1[WT] and JAK1[P832S] could induce STAT1 and STAT2 phosphorylation in a dose-dependent manner in response to interferon stimulation, despite the level of JAK1 phosphorylation remaining relatively constant (Fig. 7a,b). These observations suggest that JAK1 has a mode of function in interferon signalling that is independent of

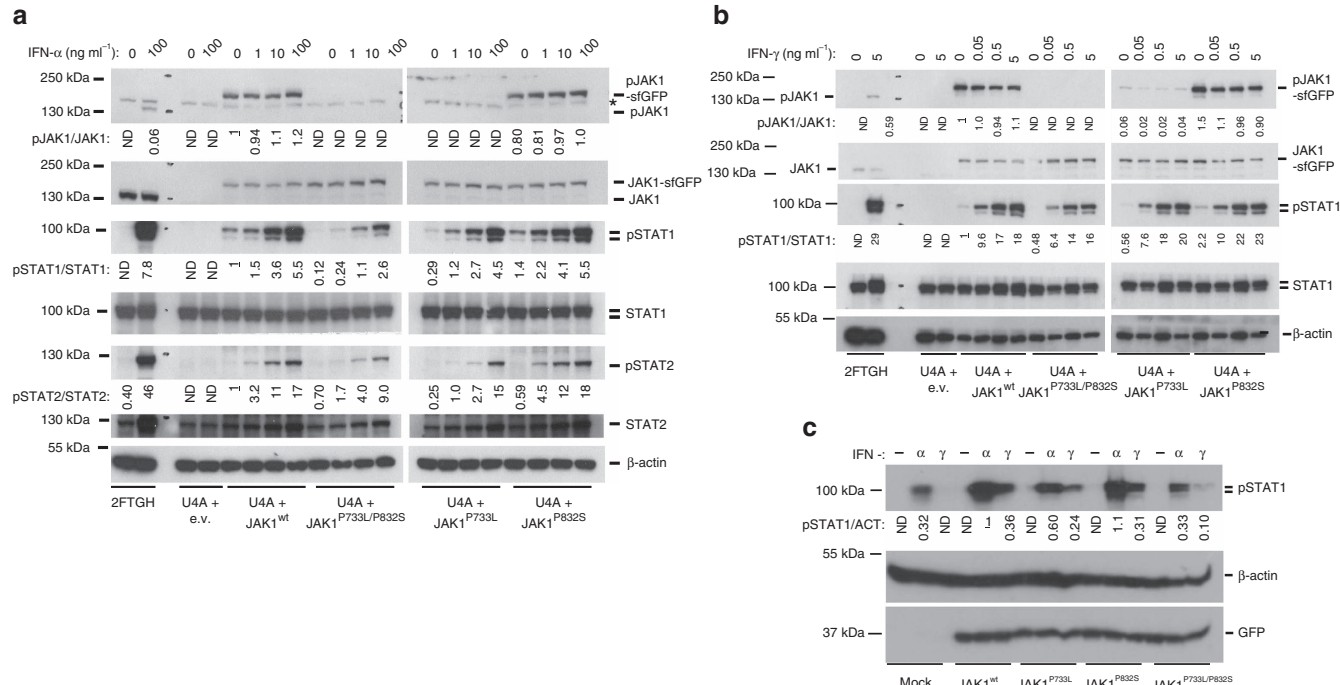

**Figure 7 | P733L mutation impairs JAK1 function. (a,b)** JAK1-deficient fibrosarcoma cells, U4A, were transfected for 24 h with vectors expressing sfGFP-fused JAK1[WT], JAK1[P733L], JAK1[P832S] or JAK1[P733L/P832S]. Then, cells were exposed to the indicated concentrations of IFN-α (**a**) or IFN-γ (**b**) for 15 min. Cells transfected with empty vector (e.v.) and parental, JAK1-competent, fibrosarcoma cells 2fTGH, served as negative and positive controls, respectively. *shows non-specific protein species detected by the antibody. Representative of four independent experiments. Fold change of band densitometry is indicated (numbers below bands and bar graphs in Supplementary Fig. 9).(**c**) Patient fibroblasts were transduced with vectors expressing JAK1[WT], JAK1[P733L], JAK1[P832S] or JAK1[P733L/P832S] and eGFP. Cells were stimulated with either IFN-α (100 ng ml$^{-1}$) or IFN-γ (1 ng ml$^{-1}$) or medium ( − ). One experiment. Fold change of band densitometry is indicated (numbers below bands).

phosphorylation of tyrosines in its activation loop, for example, providing a scaffold for the juxtaposed Janus kinase. This putative JAK1 function could explain why such a profound defect in JAK1 phosphorylation observed in the patient-derived fibroblasts leads to only modest reduction in the downstream STAT phosphorylation (Figs 3 and 5).

**JAK1 pseudokinase domain is essential in IFN-γ signalling**. To further study this hypothesis and to identify the domain responsible for kinase-independent JAK1 function, we cloned a series of its superfolder green fluorescent protein (sfGFP)-tagged kinase-dead and deletion mutants (JAK1[K908E], JAK1[KinΔ], JAK1[ΨKinΔ], JAK1[ΨKinΔ/K908E], JAK1[KinΔ/ΨKinΔ]; Supplementary Fig. 4) and studied them alongside the patient mutation (JAK1[P733L/P832S]). To ensure that the observed effects are not artifacts of JAK1 activation due to massive overexpression following cell transfection, we used JAK1-deficient Flp-In U4C cells[26], which, similarly to U4A, lack endogenous expression of JAK1, and generated cell clones that stably expressed our mutant JAK1 constructs inserted in the FRT sites. First, we used live imaging of these stable U4C cell clones to look at the subcellular localisation of the sfGFP-tagged mutant JAK1 proteins. The wild type and mutant JAK1 proteins were all similarly associated with cell membrane (Supplementary Fig. 5). Following IFN-α and IFN-γ stimulations, the phosphorylation of JAK1[P733L/P832S] and JAK1[K908E] was reduced, and phosphorylation of JAK1[ΨKinΔ] and JAK1[ΨKinΔ/K908E] was abolished (Fig. 8). After IFN-α stimulation STAT1 phosphorylation was strongly reduced in cells expressing the kinase-dead mutant JAK1[K908E] or the mutant JAK1[KinΔ] lacking the kinase domain; however, it was only slightly reduced in these

cells after IFN-γ stimulation (Fig. 8). Cells expressing the mutant JAK1[ΨKinΔ] protein lacking the pseudokinase domain showed strong reduction in phosphorylated STAT1 both after IFN-α and IFN-γ stimulations (Fig. 8). Complete abrogation of STAT1 phosphorylation was observed in cells expressing either JAK1[KinΔ/ΨKinΔ], which lacked both pseudokinase and kinase domains, or JAK1[ΨKinΔ/K908E], a kinase-dead mutant lacking also the pseudokinase domain. Taken together, these data demonstrate that JAK1 has a mode of function, which is independent of its kinase activity and its phosphorylation in the kinase domain, but requires the presence of its pseudokinase domain. This JAK1 function was particularly clear after IFN-γ stimulation: in the absence of its kinase domain, JAK1 with the functional pseudokinase domain can still transmit signalling after IFN-γ stimulation leading to STAT1 phosphorylation.

**Discussion**

Here, we describe JAK1 signalling disruptions in a patient exhibiting PID. In immune cells JAK1 mediates intracellular signalling from multiple cytokine receptors[27]. Therefore, it seems likely that impaired phosphorylation of several STAT proteins contributed to the immunodeficiency manifested by the patient. Impaired responses to IL-2 may have led to progressive T lymphopenia, whereas increased IL-6 production could have been responsible for the persistent increased serum IgG levels. JAK1-deficient mice are runted at birth and have severely reduced numbers of thymocytes, pre-B cells and mature T and B lymphocytes[28]. Although JAK1-deficient mice died perinatally, the disease in our patient has been less severe, probably because his JAK1 deficiency is partial and cells have retained ability to mediate STAT signalling. Nevertheless, the

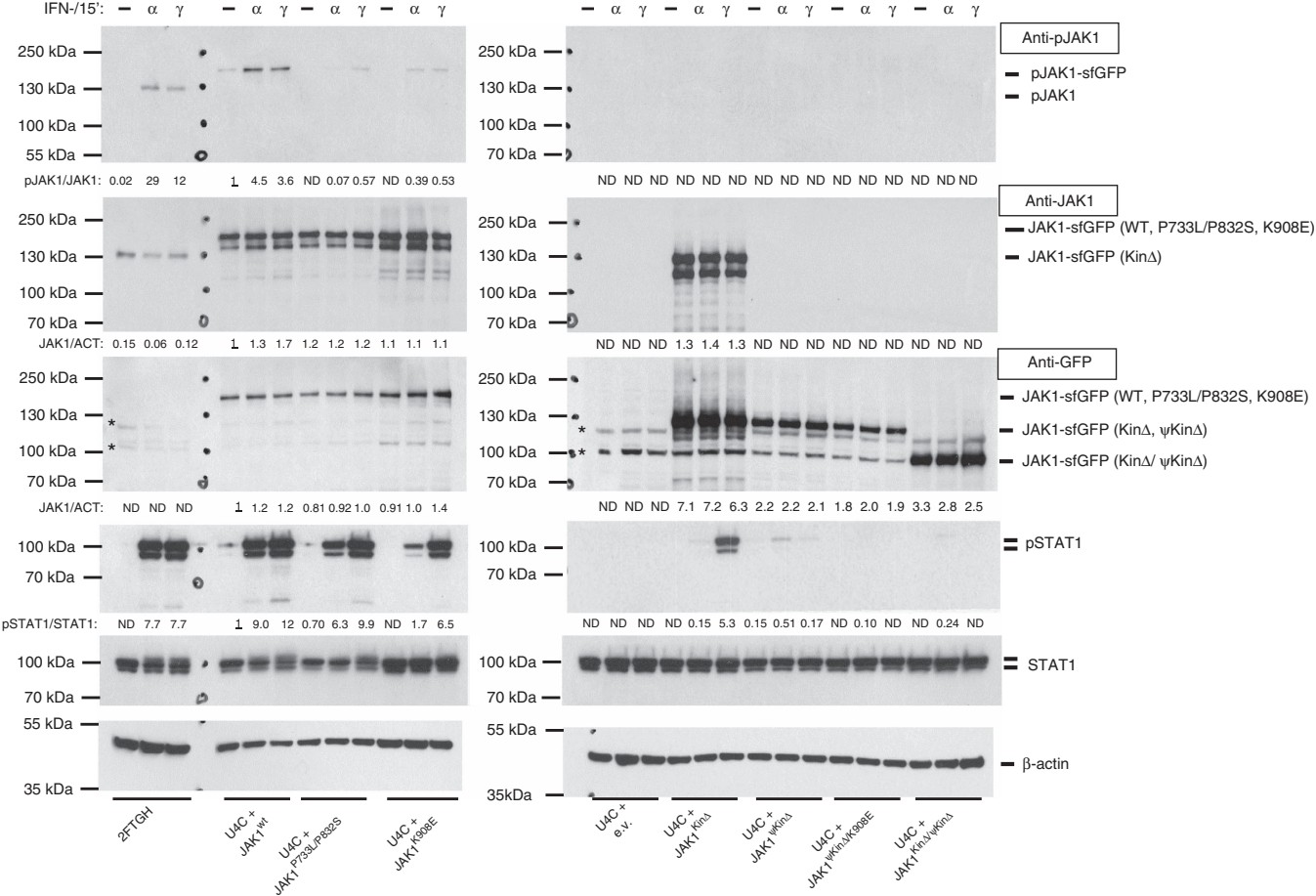

**Figure 8 | JAK1 pseudokinase domain is required for mediating IFN-γ signalling.** Stable clones of the JAK1-deficient Flp-In U4C fibrosarcoma cells, expressing JAK1$^{WT}$, JAK1$^{P733L/P832S}$, JAK1$^{K908E}$, JAK1$^{KinΔ}$, JAK1$^{ΨKinΔ}$, JAK1$^{ΨKinΔ/K908E}$ or JAK1$^{KinΔ/ΨKinΔ}$ were exposed to 100 ng ml$^{-1}$ IFN-α or 5 ng ml$^{-1}$ of IFN-γ or medium ( − ) for 15 min. Cells expressing empty vector (e.v.) and parental, JAK1-competent, fibrosarcoma cells, 2fTGH, served as negative and positive controls, respectively. *Shows non-specific protein species detected by the antibody. Note that anti-phospho-JAK1 antibody cannot be used to analyse phosphorylation of JAK1$^{KinΔ}$ or JAK1$^{KinΔ/ΨKinΔ}$, whereas anti-JAK1 antibody cannot detect JAK1$^{ΨKinΔ}$, JAK1$^{ΨKinΔ/K908E}$ or JAK1$^{KinΔ/ΨKinΔ}$, which lack relevant target epitopes. Representative of at least three independent experiments. Fold change of band densitometry is indicated (numbers below bands and bar graphs in Supplementary Fig. 10).

patient exhibited clinically significant susceptibility to atypical mycobacteria. This could be explained by a combination of T-cell lymphopenia, reduced IFN-γ production by the existing T cells, and impaired JAK1-STAT1 signalling downstream of the IFN-γ receptor. We found that hypomorphic JAK1 mutations have pleiotropic effects and affect multiple signalling pathways. Nevertheless, the disease manifested clinically with recurrent atypical mycobacterial infection, which suggests a dominant effect on the IFN-γ pathway.

Non-hematopoietic malignancy at a young age was a notable feature in our patient. Recently, somatic loss-of-function mutations in JAK1 have been associated with gynaecologic cancers[12]. Tumours also have been reported in patients with other defects in the IL-12/IFN-γ pathway, including Kaposi sarcoma[29], B cell lymphoma[30], disseminated cutaneous squamous cell carcinoma[31], oesophageal squamous cell carcinoma[32] and pineal germinoma[33]. These observations and the findings reported here demonstrate that impaired IL-12/IFN-γ signalling predisposes not only to mycobacterial infection, but also to malignancy, probably owing to impaired immune surveillance.

JAK1 inhibitors, for example, small molecule tofacitinib that inhibits JAK1 and JAK3, are used for treatment of rheumatoid arthritis and are currently tested in other

immunological disorders, such as psoriasis and inflammatory bowel disease. Serious infections, including tuberculosis, and cancers have been reported in tofacitinib clinical trials[34–37]. These observations are in line with the clinical presentation of our patient, highlighting the phenotype associated with JAK1 deficiency, either pharmacological or genetic.

Although the JAK-STAT pathway has been studied extensively, the mechanism of JAK activation upon cytokine stimulation is not entirely clear. JAKs are activated through cytokine-induced trans-phosphorylation. It was shown that at the IL-2 receptor both JAK1 and JAK3 can trans-phosphorylate each other without being phosphorylated themselves[38]. Our data show that at the IFN-γ receptor JAK1 can transmit signalling even in the absence of its kinase domain, which precludes phosphorylation of JAK2 by JAK1. Instead, our findings suggest that JAK1 pseudokinase domain is required for interaction with JAK2, and this interaction, rather than JAK1 kinase activity, is mandatory for JAK2 activation after IFN-γ stimulation leading to STAT1 phosphorylation. In contrast, IFN-α signalling requires the presence of both kinase and pseudokinase domains of JAK1 (Fig. 8). This may be explained by hierarchical trans-activation of JAKs, where upon IFN-α stimulation JAK1 first auto-phosphorylates and then phosphorylates TYK2, whereas

upon IFN-γ stimulation it is JAK2 that auto-phosphorylates first and then phosphorylates JAK1; hence, functional JAK1 deficiency markedly impairs TYK2 phosphorylation, but has no strong effect on JAK2 phosphorylation (Fig. 6). This hierarchy is consistent with previous observations that JAK2 phosphorylation does not require active JAK1, whereas JAK1 activation requires active JAK2 (ref. 39).

Although no full-length crystal structure of JAK1 or any other Janus kinase is currently available, 2D electron microscopy averages and 3D reconstructions show that JAK1 domains have conformational flexibility and that pseudokinase and kinase domains are closely associated with each other[40]. In Janus kinases the pseudokinase domain stabilises the inactive conformation of the kinase domain[41]. In structural models of JAKs, many tumour-associated activating mutations map to the interface between the pseudokinase and kinase domains, presumably disrupting interaction between them, which facilitates activation of the kinase domain[18,41]. For example, an activating mutation F734L in JAK1, found in a sample from a T-cell acute lymphoblastic leukaemia patient[42,43], maps in the β7–β8 loop of the pseudokinase domain that interacts with the β2–β3 loop of the kinase domain[18,41]. The P733L mutation that we found in our patient maps in the same pseudokinase β7–β8 loop next to F734L, however, as we have shown here, P733L reduces JAK1-mediated signalling, suggesting that it may enhance auto-inhibitory interaction between the pseudokinase and the kinase domains.

In summary, hypomorphic recessive germline JAK1 mutations that affect multiple signalling pathways were found in a PID case that manifested with atypical mycobacterial infections and increased susceptibility to cancer. This phenotype associated with a long-term functional JAK1 deficiency predicts effects of prolonged administration of JAK1 inhibitors. We also describe a mechanism of JAK1 function in interferon signalling, which is independent of phosphorylation of tyrosines in its activation loop and its kinase function. These findings illustrate that discovery of novel naturally occurring mutations can reveal the molecular basis for human disorders, and also helps to understand fundamental biological mechanisms, even in well-characterized pathways such as the JAK-STAT pathway.

## Methods

**Ethics statement.** Blood samples and skin biopsies were obtained with informed consent from all subjects in accordance with the Declaration of Helsinki and with approval from the ethics committee (NRES Committee London—Bloomsbury 06/Q0508/16).

**Whole-exome sequencing.** Library preparation, exome capture and sequencing have been done according to the manufacturers' instructions. For exome target enrichment Agilent SureSelect 38 Mb kit was used. Sequencing was done using Illumina HiSeq with 94 bp paired-end reads. Reads from raw FASTQ files were aligned to the hg19 reference genome using Novoalign version 2.08.03. Duplicate reads were marked using Picard tools MarkDuplicates. Calling was performed using the haplotype caller module of GATK (https://www.broadinstitute.org/gatk), creating genomic variant call format (gVCF)-formatted files for each sample. The individual gVCF files were combined into gVCF files containing 100 samples each. The final variant calling was performed using the GATK 'GenotypegVCFs' module jointly for all cases and controls. Variants quality scores were then re-calibrated according to GATK best practices separately for indels and SNPs. Resulting variants were annotated using software ANNOVAR.

**Sequence alignment and protein modelling.** The known JAK1 protein sequences from Ensembl (http://www.ensembl.org) were aligned using the Multiple Sequence Alignment ClustalW2 (http://www.ebi.ac.uk/Tools/msa/clustalw2/). Modelling of the pseudokinase (JH2) and kinase (JH1) domains of JAK1 was done by I-TASSER[44], using JAK1 sequence (AA 583–1153) and the three-dimensional structure of TYK2 (PDB 4OLI) as a template. The model with the best confidence score (C-Score) was visualized by Visual Molecular Dynamics software (http://www.ks.uiuc.edu/Research/vmd/).

**Cells.** Cell lines U4A and 2fTGH were kindly provided by Dr George Stark. Flp-In U4C (U4C-FRT) cells were kindly provided by Dr Claude Haan. Lack of JAK1 expression in JAK1-deficient U4A and U4C cells has been verified by western blotting. Cells were tested for mycoplasma contamination and were shown to be mycoplasma-negative. The parental 2fTGH human fibrosarcoma cell line and U4A and U4C-FRT cell lines, patient and control fibroblasts as well as 293T cells were cultured in DMEM medium (E15–810,PAA) and 10% heat-inactivated FCS fetal bovine serum (35 5500, Lot 001514, BD, Cowley, Oxford). Parental Flp-In U4C (U4C-FRT) cells or JAK1-complemented stable clones were kept in the presence of 100 μg ml$^{-1}$ Zeocin (Invivogen) or 250 μg ml$^{-1}$ Hygromycin B Gold (Invivogen), respectively.

**Determination of mRNA levels by relative real-time PCR.** Total RNA was extracted using RNAeasy Plus Mini kit (Qiagen) from fibroblast left unstimulated or stimulated with 1–100 ng ml$^{-1}$ IFN-α for 15 h or 0.01–1 ng ml$^{-1}$ IFN-γ for 8 h. 0.3–1 μg RNAs were then reverse-transcribed with Oligo-dT using RevertAid Reverse Transcriptase (ThermoFisher). For the determination of the mRNA levels of the target genes, we performed qPCR using gene specific primers (Supplementary Table 2) and KAPA SYBR FAST qPCR (KK4602, Kapa biosystems). Fold changes were calculated using the ΔΔCT method against house-keeping genes.

**Cloning, mutagenesis and transfection.** For cloning purposes *hJAK1* gene was cloned into the vector pcDNA3.1 (+) (Invitrogen), used as a template for PCR and subcloned into a pHR-Ub-Em vector (4), using standard cloning procedures. Myc-tagged or sfGFP-tagged JAK1 constructs were subsequently cloned in pcDNA6A-myc-His (ThermoFisher, ref. V22120). Cloning of JAK1-sfGFP cDNAs in pcDNA5/FRT/TO (ThermoFisher, V652020) was performed via ApaI/KpnI sites (JAK1-sfGFP in pcDNA6A as donor plasmids). The mutations P733L, P832S, P733L/P832S and K908E were introduced using QuikChange Site-Directed mutagenesis strategy. JAK1 deletion mutants JAK1$^{\Psi Kin\Delta}$ (del aa. 583–855), JAK1$^{Kin\Delta}$ (del aa. 875–1153) and JAK1$^{\Psi Kin\Delta/Kin\Delta}$ (del aa. 583–1153) were generated via Golden Gate assembly, with outward primers carrying BsmbI (New England Biolabs) sites. Transient transfections of U4A cells were carried out using Lipofectamine reagent (ThermoFisher), following manufacturer's instructions. Stable clones of complemented U4C-FRT were generated by contrasfecting the proper JAK1-sfGFP in pcDNA5/FRT/TO construct along with pOG44 (ThermoFisher), followed by selection in Hygromycin-containing medium. The primers used in the cloning and mutagenesis are summarized in the Supplementary Table 2. sfGFP is a genetically modified form of GFP with reduced dimerisation properties in comparison with the wild-type GFP[45].

**Lentivirus preparation and transductions.** The lentivirus stocks were prepared by transient transfection of 293T cells (75 cm$^2$ flasks at 50% confluency) with the envelope plasmid pMD.G (4 μg), the packing plasmid CMV8.91 (4 μg) and the expression plasmid pHR-UbEm (6 μg) along with 35 μl the transfection reagent Transit 2020 (5454, Mirus) following the manufacture instructions. 24 h post transfection (hpt) the media was replaced and medium was harvested at 48 and 72 h post transfection, pooled, cleared by low-speed centrifugation (1,200 rpm, 5 min), and filtered through 0.45-μm-pore-size filters, and titered by limited dilution scoring for eGFP-positive cells 3–5 days after infection. Virus stocks were stored up to 3 weeks at 4 °C for transduction experiments. Transductions of patient fibroblast and U4A cells were carried out by infection at a multiplicity of infection of 1–3 in the presence of 8 μg ml$^{-1}$ of Polybrene (107689, Sigma) overnight, then the virus containing media was replaced by fresh media, samples were stimulated and harvested 2–5 days after infection.

**Cells stimulations and immunoblotting.** STATs' phosphorylation was tested in complemented U4A cells, U4C-FRT cells or patient fibroblasts along with appropriate control cells upon stimulation with IFN-α (11101–2, PBL) or IFN-γ (IF002, Millipore), in full growth medium at 37 °C. Then, the cells were trypsinized, washed and lysed in cold radioimmunoprecipitation assay buffer supplemented with phosphatases (P5726, Sigma) and protease inhibitors (11836153001, Roche) for 30 min at 4 °C. The lyses were then centrifuged at 10,000*g* for 10 min at 4 °C, the supernatants were quantified and 10–50 μg of total protein was separated by 10% SDS–PAGE and analysed by western blot. Membranes were cut horizontally according to molecular size markers, and stripes were incubated with different Abs. Immunoblots were developed with the enhanced chemiluminescence western blotting Reagent (Amersham). The following Abs were used: anti-JAK1 (610231, BD Biosciences; 1/1,000 dilution), anti-pJAK1 Tyr$^{1022/1023}$ (3331, Cell Signaling Technology; 1/1,000 dilution), anti-JAK2 (Clone D2E12, 3230, Cell Signaling Technology; 1/1,000), anti-pJAK2 Tyr$^{1007/1008}$ (Clone C80C, 33776, Cell Signaling Technology; 1:1,000), anti-TYK2 (Clone D4I5T, 14193, Cell Signaling Technology; 1:1,000), anti-pTYK2 Tyr$^{1054/1055}$ (Clone C80C, 9321, Cell Signaling Technology; 1:1,000), anti-STAT1 (9172, Cell Signaling Technology; 1:1,000), anti-pSTAT1 Tyr$^{701}$ (Clone 58D6, 9167, Cell Signaling Technology; 1:1,000), anti-STAT2 (4594, Cell Signaling Technology; 1:1,000), anti-pSTAT2 Tyr$^{689}$ (07–224, Millipore; 1:2,000), anti-β-Actin (Clone AC-15, A5451, Sigma-Aldrich; 1:10,000), anti-GFP (11814460001,

Roche or ab290, AbCam; 1:2,000) and anti-Myc (clone 9B11, 2276, Cell Signaling Technology; 1:1,000). Band densitometry was determined by Image Studio Lite (Licor). Fold changes of phospho/non-phosphorylated proteins were calculated against the indicated internal reference sample. Similarly, JAK1 protein levels in primary fibroblasts were determined after normalisation with β-actin (loading control). Uncropped scans of western blots are shown in Supplementary Fig. 11.

**Microscopy.** U4C cells complemented with sfGFP-tagged JAK1 mutants were seeded at 50,000 cell per well on 24-well Sensoplates (662892, Greiner). Two days later, living cells were rinsed twice in 1xPBS and imaged in Live Cell Imaging Solution (A14291DJ, ThermoFisher). Nuclei were stained with NucBlue reagent (R37605, ThermoFisher). Micrographs were acquired at a Leica SP5 confocal microscope.

**FACS analysis of STAT phosphorylation.** Blood was collected in EDTA. In total, $100 \mu l$ of blood was left unstimulated and $100 \mu l$ was stimulated with indicated cytokines for 10 min (IL-2, Chiron, $10^5 \, U \, ml^{-1}$; IL-4, ImmunoTools, $1 \, \mu g \, ml^{-1}$; IL-6, R&D systems, Abingdon, England, $100 \, ng \, ml^{-1}$; IL-10, R&D systems, $500 \, ng \, ml^{-1}$; IFNγ, R&D systems, $200 \, ng \, ml^{-1}$; IFNα, PBL Interferon Source, $10^6 \, U \, ml^{-1}$; IL-27, R&D systems, $500 \, ng \, ml^{-1}$). Red cells were lysed and phosphorylation state fixed using Lyse/Fix (BD Biosciences). Cells were permeabilised with Perm Buffer III (BD Biosciences) before being stained with $5 \, \mu l$ surface antibodies: APC Anti-CD3 1:20 dilution, or PerCP Anti-CD4 1:20, and PE anti-STAT1 (p701) 1:20, or Alexa Fluor 488 Anti-STAT5 (Y694) 1:20, or Alexa Fluor 488 Anti-STAT4 (Y693) 1:20, or Alexa Fluor 488 Anti-STAT3 (49/STAT3) 1:20, or Alexa Fluor 488 Anti-STAT6 (Y641) 1:20 (all from BD Biosciences). Gating was done on the following lymphocyte populations: total lymphocytes for the analysis of STAT1 phosphorylation after IFN-α stimulation; CD3- lymphocytes for the analysis of STAT1 phosphorylation after IFN-γ stimulation and of STAT3 phosphorylation after IL-10 or IL-6 stimulations; CD3+ lymphocytes for the analysis of STAT4 phosphorylation after IFN-α stimulation, of STAT5 phosphorylation after IL-2 stimulation and of STAT6 phosphorylation after IL-4 stimulation; CD4+ lymphocytes for the analysis of STAT1 phosphorylation after IL-27 stimulation. The stained cells were detected using a FACsCalibur (BD Biosciences); 10,000 gated events were collected. Analysis was performed using CellQuest software (Becton Dickinson).

**Whole-blood cytokine production assays.** Whole blood was diluted 1:5 in RPMI into 96-well F plates (Corning) and activated by single stimulation with IL-12 ($20 \, ng \, ml^{-1}$; R&D Systems), PHA ($10 \, \mu g \, ml^{-1}$; Sigma-Aldrich), LPS ($1 \, \mu g \, ml^{-1}$) List Biochemicals, IFN-γ ($2 \times 10 \, IU \, ml^{-1}$, Imukin, Boehringer Ingelheim), IFN-α ($2 \times 10^3 \, IU \, ml^{-1}$, Intron A, Schering Plough, UK) or using co-stimulations as indicated. Supernatants were taken at 24 h. Cytokines were measured using standard ELISA according to the manufacturer's recommendations (IFN-γ, Pelikine, Sanquin, NL), or multiplexed (TNFα, IL-12, IL-10, IL-6, R + D Systems Fluorokinemap) on a Luminex analyser (Bio-Plex, Bio-Rad, UK). Data were statistically analysed by the two-tailed Mann–Whitney test using Prism 6 (GraphPad Software).

**Data availability.** The data that support the findings of this study are available from the corresponding author upon request.

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

## Acknowledgements

S.N. is a Wellcome Trust Senior Research Fellow in Basic Biomedical Science (095198/Z/10/Z). The work was also supported by the Higher Education Funding Council for England (S.O.B.), the Wellcome Trust (090233/Z/09/Z, A.J.T.), Great Ormond Street Hospital Children's Charity (F.H., K.N.), the National Institute for Health Research (NIHR) Great Ormond Street Hospital Biomedical Research Centre (K.C.G.), the NIHR Cambridge Biomedical Research Centre (S.N., D.K., and R.D.) and the Alfonso Martin Escudero Foundation (V.D.-C). Cell lines U4A and 2fTGH were kindly provided by Dr George Stark. Flp-In U4C (U4C-FRT) cells were kindly provided by Dr Claude Haan. We thank Dr Luke Thomas for his assistance with live cells imaging. We also thank Simon McCallum from the Cambridge NIHR BRC Cell Phenotyping Hub.

## Author contributions

D.E. performed analyses of primary fibroblasts and cell lines and the structural protein analysis. I.A., F.H., M.G., K.N., V.D.-C. contributed to these analyses. S.O.B. and A.J.T. treated the patient and collected clinical data and samples. K.C.G. performed fluorescence-activated cell sorting analyses. R.D. and D.K. performed whole blood assays. V.P. and S.N. analysed exome data. J.C. performed sequencing. S.N., A.J.T. and S.O.B. supervised the study. S.N. wrote the first draft; all authors contributed to the writing of the manuscript.

## Additional information

Competing financial interests: The authors declare no competing financial interests.

