## [Peer Review File · Nature Communications]

Reviewers' comments:

Reviewer #1 (Remarks to the Author):

Eletto et al studied one patient from consanguineous parents who presented at early age with repeated atypical mycobacteria infection. At the age of 21 the patient showed mild T cell lymphopenia and died at the age of 23 of metastatic bladder carcinoma. Two homozygous mutations (P7333L and P832S) in JAK1 were identified by exome sequencing. The two mutated residues are located in a regulatory portion of the protein, the pseudokinase domain. Both parents carry the two mutations but in heterozygote state. This is a potentially interesting finding.

Cytofluorimetric analysis of phospho-STATs in PBMC lead the authors to conclude on a rather broad defect in the response to several cytokines. Somehow surprisingly, IL-6 signaling was not impaired. Cytokine secretion was measured in whole blood stimulation with PHA or LPS.

The authors measured signaling to IFN α and IFN γ by western blots in patient fibroblasts and in JAK1-minus U4A cells transiently transfected with single and double JAK1 mutants. The single P7333L mutant is dysfunctional as non phosphorylated and yet signaling is not much affected. No measure of catalytic activity is provided.

Several JAK1 truncation mutants stably transfected in U4A cells were analyzed for IFN α vs IFN γ .

The major conclusions of the study is that the hypomorphic JAK1 is causative to the mycobacterial susceptibility and the development of the cancer in the patient. I do not think that these claims are substantiated by the analyses shown.

Western blot analyses and the only analysis of gene induction (suppl Fig. 3) were performed with exceedingly elevated doses of cytokines. Experiments shown in Figure 6 do not provide any novel information on the role of JAK1 in IFNs signaling, nor on the defect of the mutants identified by the authors.

The discussion is not particularly deep. The link to cancer remains totally hypothetical.

Figure 1C

The FERM domain encompasses most of the N region and is closely juxtaposed to the SH2-like domain. PTK-pseudokinase domain and PTK-kinase could be more simply denoted as

pseudokinase domain and TK domain

Concerning the structure shown, the source should be cited in the legend.

Figure 2a

PBMC from controls and the patient were stimulated for 10 min with a single (highly saturating) dose of cytokine, and the percentage of cells positive for phospho-STATs was measured by FACS. A reduced level of STAT activation was observed in patient cells in all stimulation conditions, with the exception of IL-6 (P-STAT3).

Assays were normalized for T cell count. Gating is described in Mat&Meth but not shown. The age of the patient when blood was drawn is not mentioned. Given the lymphopenia that the patient slowly developed, this is an important point. The origin of the many healthy controls is not specified.

Stimulation was on PBMC, as written in the text and legend, or on whole blood as described in Mat&Meth (page 21, line 460)? Time of stimulation should be mentioned in the legend.

Figure 2b-f

Whole blood was challenged for 24hr with either PHA or LPS and cytokine production (IFN γ , IL-10, TNF α , IL-6, IL-12) was measured by Elisa or multiplex. Are LPS responses increased ?

Figure 3: The authors conclude that the mutant JAK1 protein is expressed at similar levels in patient and in control fibroblasts. This is not what is shown in panels a and c (and Suppl Fig. 2a), where JAK1 appears less abundant in patient cells than control..

Again blots are overexposed and no quantification is shown.

Have the authors measured TYK2 and JAK2 phosphorylation in these conditions ? are they reduced as well ?

At a very high dose of IFN α (100ng/ml), a defect in JAK1 phosphorylation is clearly observed in patient fibroblasts. Yet, the effect on STAT1 Pi is rather small.

To conclude on the consequence of defective JAK1 on STAT1 phosphorylation, 100x less cytokine should be used to measure P-STAT.

Doses of IFN α are in some experiments in ng/ml and in others in IU/ml. This should be consistent.

The level of total STAT1 at 15min stimulation seems higher. Explain.

Again blots are much too exposed.

Figure 4: here the authors conclude that TYK2 phosphorylation in response to IFN α is more affected than JAK2 phosphorylation in response to IFN γ . This comparison is not tenable.

Different stimulating cytokines (for which level of cognate receptors likely differs) at non comparable doses were used. Affinity of phospho-specific antibodies can greatly vary. The amount of phosphorylated JAK1 in the two conditions is not shown.

Figure 5: here the authors transiently transfected U4C cells with GFP fusion of JAK1 wt or mutants. The basal level of phosphorylation of JAK1 in transfected is expectedly high and possibly also further increased by GFP-mediated dimerization.

Quite convincingly JAK1 P7333L is not phosphorylated, neither basally nor after cytokine stimulation and yet STAT1 phosphorylation in these transfected cells is very weakly affected. Stable clones should be analyzed.

Pag 8, line 147, the particular location of the 2 Proline residues within the pseudokinase domain should be more finely described.

Page 10, line 202: the mutant variant ?

Suppl Fig. 2a and Fig 3: The level of JAK1 in patient's fibroblats seems definitely lower than in the two controls, as opposed to what the authors conclude. This is also seen in Fig. 3.

All JAK1 blots are overexposed. Quantification relative to actin should be presented in less exposed blots or in westerns with lower amount of lysate.

Line 227 It cannot be concluded that the mutation P7333 "has an inhibitory effect" is not appropriate

Suppl. Fig. 3: time of stimulation is not mentioned.

Pag 22, line 485: the second IFN γ in that line should read IFN α (Intron

Reviewer #2 (Remarks to the Author):

This manuscript from Eletto and colleagues describes an immunodeficient patient, highly susceptible to mycobacterial infection, who developed an early onset metastatic bladder cancer. Whole exome sequencing identified 5 homozygous rare variants, including two missense mutations in the JAK1 gene, resulting in the amino acid changes p.P733L and p.P832S in JAK1's pseudokinase domain. The authors focus on these two JAK1 mutations and characterise JAK1-mediated signalling in patient cells, HEK293T cells and in human fibrosarcoma cell lines lacking endogenous JAK1 (U4A, Flp-In U4C cell lines).

The authors dismiss the possibility of any contribution to patient's condition from the other three genes identified with homozygous rare variants, which could potentially play a role. For example, one of these genes, ANXA1 (annexin A1) (Suppl Table 1), is involved in innate and

adaptive immunity, as well as potential tumour-suppressor activity. Annexin A1 can also be linked to IFN-g production, as this is increased in a knock out mouse model upon OVA stimulation (Yang, J. Immun., 2013). While it seems reasonable to associate JAK1 mutations to primary immunodeficiency (PID) via IFN-g signalling, as there is literature linking mutations in IFNgR1 and IFNgR2 to PID, the link to cancer susceptibility is more tenuous. Unfortunately this manuscript lacks an in vivo model that would convincingly demonstrate that JAK1 P733L is driving the diseases seen in the patient, and this would be necessary to support the conclusions made by the authors.

No attempt is made to discuss the effects of these amino acid changes at a structural level and how this might alter JAK1 function. It is not clear why the authors have chosen to show a homology-based model from 2008 (fig 1c), itself based on a prediction of JAK2, when crystal structures of JAK1 JH2 and JH1 are publically available. It would perhaps be more useful to map the mutations to the JH2 structure alone (PDB 4L00), or to model JAK1 JH1 and JH2 based the only crystal structure of a JAK to date with both kinase and pseudokinase domains, that of TYK2 (PDB 4OLI). Although no full length crystal structures of JAK1 (or any other JAK) are currently available, there is some cryo-EM work on full length JAK1 (Lupardus, 2011) which is relevant to any discussion of JAK1 domain configuration.

Other points

- In figure 2 it is not clear how many healthy controls were tested, or whether 1 healthy control was simply tested several times. It would be useful to have an idea of how much variation exists in a healthy population to be able to decide whether signalling defects exist in the patient's cells.
- pSTAT1 in U4A JAK1 P733L cells following IFN-g stimulation doesn't appear to be reduced compared to U4A JAK1 WT cells (Fig 5b), as claimed in the text (lines 230-231); if anything, it appears to be increased.
- Molecular weight markers could be useful for the lower portion of fig. 6.
- Also in Fig. 6, the results for the JAK1 kinase and pseudokinase domain deletion constructs are interesting, but is it possible to show a larger section of the Phospho-JAK1 blot that includes the region where you would expect to see JAK1 Ψ Kin Δ and JAK1 Ψ Kin Δ /K908E? Additionally, the hierarchical transactivation model proposed here would be better supported with western blots using antibodies raised against the phosphorylated activation loops of JAK2 and TYK2.
- Even though the authors point out that over expression of JAKs are known to result in constitutive phosphorylation of the activation loop, it is not clear how using the U4C cell line would get around this, and indeed phosphorylation of JAK1 and STAT1 in the absence of cytokine are observed in U4C JAK1 WT, and to a lesser degree in U4C JAK1 P733L/P832S (Fig.6).

Reviewer #3 (Remarks to the Author):

This is a clearly written manuscript documenting a novel PID caused by homozygous mutation in JAK1.

.The results are interesting and of great importance

The study is very well conducted and the conclusions are justified

The novelty resides in the novel gene described as well as in defining a signaling function of JAK1 independent of its kinase activity

There are few concerns to address

1. Given that the biochemical changes are expected to increase susceptibility to Mycobacteria the authors make only scant reference to the Mycobacterial infection of the patient. A better documentation is needed
2. While the representative gels in the Figures are quite clear the manuscript will be stronger if the results of the replicate experiments are shown in graphical quantitative plots next to the representative gels . This should be done at least for the major findings of decreased INF γ and INF α driven JAK1 and INF γ phosphorylation
3. The data on gene induction in Supplemental Fig 3 is important and should be shown as a standard Figure

Responses to Comments

Reviewer #1:

1. “The authors measured signaling to IFN α and IFN γ by western blots in patient fibroblasts and in JAK1-minus U4A cells transiently transfected with single and double JAK1 mutants. The single P733L mutant is dysfunctional as non phosphorylated and yet signaling is not much affected. “

Indeed, we found that a rather dramatic reduction of phosphorylation of the mutant JAK1 leads to only partially reduced downstream signalling (e.g. phosphorylation of STAT1). This discrepancy suggests that JAK1-mediated mechanisms of signal transduction are only partially dependant on its phosphorylation. We discuss this finding in the manuscript.

2. “No measure of catalytic activity is provided.”

We measured kinase activity of JAK1 by studying STAT phosphorylation in primary fibroblasts and lymphocytes, as well as U4A and U4C cell lines. We now quantified protein phosphorylation in all experiments and present quantification results together with Western blot scans.

3. “The major conclusions of the study is that the hypomorphic JAK1 is causative to the mycobacterial susceptibility and the development of the cancer in the patient. I do not think that these claims are substantiated by the analyses shown.”

We show that JAK1 mutations affect IFN signalling, leading to impaired STAT activation, including that of STAT1. It has been firmly established that mutations in this pathway, and specifically hypomorphic STAT1 mutations, lead to primary immunodeficiencies that manifest with susceptibility to mycobacterial infection (OMIM 614892 and 613796). Therefore, we believe that the causative link between JAK1 mutations and mycobacterial infection in our patient is quite clear. The link between the primary immunodeficiency caused by JAK1 mutations and cancer diagnosed in the patient is less certain. Thus, we carefully avoid suggesting a causative link between them (although we think that it is important to mention bladder carcinoma, as an unusual early feature in the patient):

Title: *“Bi-allelic JAK1 mutations in immunodeficient patient with mycobacterial infection and cancer”*

Abstract: *“Our findings clarify mechanisms of JAK1 signaling and demonstrate its critical role in protection against mycobacterial infection and, possibly, in immunological surveillance of cancer.”*

Introduction: *“...JAK1 mutation that leads to functional JAK1 deficiency associated with susceptibility to atypical mycobacterial infection and early-onset bladder carcinoma”*

Also please see our response to Reviewer #2 comment 2 below.

4. “Western blot analyses and the only analysis of gene induction (suppl Fig. 3) were performed with exceedingly elevated doses of cytokines.”

In Figures 3, 5, 6 and 7 we present Western blots after stimulation with different cytokine doses ranging 100 fold.

We now significantly expanded our analysis of gene expression induction using various doses of cytokines. The results are now shown in Figure 4 and Supplementary Fig. 3.

5. “Experiments shown in Figure 6 do not provide any novel information on the role of JAK1 in IFNs signaling, nor on the defect of the mutants identified by the authors.”

In Figure 6 (now Fig 8) we studied stable clones of the JAK1-deficient U4C cells complemented with various JAK1 mutants. These cells did not show such a pronounced induction of the signalling cascade by mass activation as we observed in transiently transfected U4A (Fig 7). Therefore, the stable clones provide a valuable addition to our experiments.

6. “The discussion is not particularly deep.”

We now extended Discussion by addressing potential effects of mutations on the JAK1 structure and functions.

7. “The link to cancer remains totally hypothetical.”

Please see our response to comment 3 above and response to Reviewer #2 comment 2 below.

8. “Figure 1C

**The FERM domain encompasses most of the N region and is closely juxtaposed to the SH2-like domain. PTK-pseudokinase domain and PTK-kinase could be more simply denoted as pseudokinase domain and TK domain
Concerning the structure shown, the source should be cited in the legend.”**

We corrected the linear diagram and added coordinates of the JAK1 domains. JAK1 pseudokinase and kinase domain structure (Fig 1d) was modelled using the published structure of TYK2 (PDB 4OLI). This information is now in the figure legend.

9. “Figure 2a

**PBMC from controls and the patient were stimulated for 10 min with a single (highly saturating) dose of cytokine, and the percentage of cells positive for phospho-STATs was measured by FACS. A reduced level of STAT activation was observed in patient cells in all stimulation conditions, with the exception of IL-6 (P-STAT3). Assays were normalized for T cell count.
Gating is described in Mat&Meth but not shown.”**

We now show gating for pSTAT1 after IFN- α and IFN- γ stimulation in Fig 2a, right panel.

10. “The age of the patient when blood was drawn is not mentioned. Given the lymphopenia that the patient slowly developed, this is an important point.”

The FACS analyses were done when the patient was 20 years old. We have now included age of the patient in the legend of Figure 2.

11. “The origin of the many healthy controls is not specified.”

We studied healthy travel controls whose blood sample was collected simultaneously and shipped to the laboratory together with the patient’s sample.

12. “Stimulation was on PBMC, as written in the text and legend, or on whole blood as described in Mat&Meth (page 21, line 460)?”

Stimulation was done on whole blood, as described in Methods. We corrected it in the text and legend.

13. “Time of stimulation should be mentioned in the legend.”

We now included time of stimulation in the legend.

14. “Figure 2b-f

Whole blood was challenged for 24hr with either PHA or LPS and cytokine production (IFN γ , IL-10, TNF α , IL-6, IL-12) was measured by Elisa or multiplex. Are LPS responses increased ?”

After LPS stimulation production of IL-6 and TNF α was increased, while production of IFN γ and IL-12 was normal.

15. “Figure 3:

The authors conclude that the mutant JAK1 protein is expressed at similar levels in patient and in control fibroblasts. This is not what is shown in panels a and c (and Suppl Fig. 2a), where JAK1 appears less abundant in patient cells than control..”

We now repeated this analysis three times, comparing levels of JAK1 expression in fibroblasts of the patient and two healthy controls in 3 biological replicates and quantified the results. Indeed, the level of JAK1 expression seems reduced, but this reduction is small, especially in comparison with one of the controls. Such a small reduction cannot account for the massive functional deficiencies (e.g. in the level of phospho-JAK1 upon exposure to IFNs). Still, to account for the difference in the JAK1 level, in other Figures we present phospho-JAK1 results as a ratio of phospho-JAK1/JAK1.

We re-phrased in the manuscript:

“The patient-derived variant of JAK1 was expressed at a slightly lower level than the wild-type JAK1.” and added new data and quantification in Suppl Fig 2.

16. Figure 3:

“Again blots are overexposed and no quantification is shown.”

We now present less exposed blots with added quantification.

17. “Have the authors measured TYK2 and JAK2 phosphorylation in these conditions ? are they reduced as well ?”

Yes, we studied levels of phosphorylated TYK2 and JAK2 after stimulation with different concentrations of IFN α and IFN γ ; the results are shown in Fig 6.

18. “At a very high dose of IFN α (100ng/ml), a defect in JAK1 phosphorylation is clearly observed in patient fibroblasts. Yet, the effect on STAT1 Pi is rather small.”

This observation is one of the findings of our study; we suggest that it is explained by the structural role of JAK1 in signal transduction that is independent of its phosphorylation:

“These observations suggest that JAK1 has a mode of function in interferon signalling that is independent of its phosphorylation. This putative JAK1 function would explain why such a profound defect in JAK1 phosphorylation observed in the patient-derived fibroblasts leads to only modest reduction in the downstream STAT phosphorylation (Fig. 3).”

19. “To conclude on the consequence of defective JAK1 on STAT1 phosphorylation, 100x less cytokine should be used to measure P-STAT.”

In Fig 3a and 3b we used different cytokine concentrations ranging 100 fold, i.e. 1, 10, 100 ng/ml of IFN α and 0.01, 0.1, 1 ng/ml of IFN γ . In addition, in Fig 5a and 5b we used different cytokine stimulation times (1, 5, 15 min , 4 h, 24 h). Here we on purpose used the highest cytokine concentrations, i.e. 100 ng/ml of IFN α or 1 ng/ml of IFN γ , and we still can observe remarkable differences in JAK1 phosphorylation

between the patient and the control cells, demonstrating that even strong stimulations cannot overcome functional JAK1 deficiency in the patient cells.

20. “Doses of IFN α are in some experiments in ng/ml and in others in IU/ml. This should be consistent.”

Corrected

21. “The level of total STAT1 at 15min stimulation seems higher. Explain.”

This was an artefact of western blotting. We repeated this analysis, please see Fig 5a.

22. “Again blots are much too exposed.”

We now present less exposed blots

23. “Figure 4: here the authors conclude that TYK2 phosphorylation in response to IFN α is more affected than JAK2 phosphorylation in response to IFN γ . This comparison is not tenable. Different stimulating cytokines (for which level of cognate receptors likely differs) at non comparable doses were used. Affinity of phospho-specific antibodies can greatly vary.”

Of course, doses of IFN γ and IFN α used for stimulation, levels of receptors and affinity of antibodies differ between these cytokines. Therefore, we do not compare them directly. Instead, we study to what extent the responses to IFN α or IFN γ are affected in the patient relative to control cells stimulated with the same cytokine. In Fig 4 (now Fig 6) we can clearly see that in response to IFN α stimulation the level of phospho-TYK2 is strongly reduced in the patient relative to control, while the level of phospho-JAK2 is only slightly reduced in the patient relative to control after IFN γ stimulation (please see graphs in Fig 6) .

24. “The amount of phosphorylated JAK1 in the two conditions is not shown.”

We now added both phospho-JAK1 and JAK1 in this Figure.

25. “Figure 5: here the authors transiently transfected U4C cells with GFP fusion of JAK1 wt or mutants. The basal level of phosphorylation of JAK1 in transfected is expectedly high and possibly also further increased by GFP-mediated dimerization.”

This is now Fig 7. Here we used genetically modified form of GFP, known as superfolder GFP (sfGFP), which has reduced dimerization properties in comparison to GFP (Pédrelacq et al, Nat Biotechnol. 2006) . We clarified this in the figure and in the text.

Furthermore, in some of our additional transfection experiments we used JAK1 tagged with myc, instead of sfGFP, and observed similar effects on the basal levels of phosphorylation of the wild type JAK1 and lack of phosphorylation of the patient (P733L/P832S) JAK1 mutant or the K908E JAK1 mutant. Therefore, the effect of the mutations on JAK1 phosphorylation is independent of the JAK1 tag.

26. “Quite convincingly JAK1 P7333L is not phosphorylated, neither basally nor after cytokine stimulation and yet STAT1 phosphorylation in these transfected cells is very weakly affected. Stable clones should be analyzed.”

We analysed stable clones; the results are shown in Figure 8.

27. “Page 8, line 147, the particular location of the 2 Proline residues within the pseudokinase domain should be more finely described.”

We have done additional structural analysis shown in Fig 1d and now added in Results:

“Similarly to other Janus kinases, JAK1 has FERM and SH2 domains, which are responsible for interaction with the cytokine receptor, pseudokinase (JH2) domain, which regulates kinase activity, and kinase (JH1) domain^{16,17}. The proline residues at JAK1 positions 733 and 832 are located in the pseudokinase domain (Fig. 1c). They are conserved within the human Janus kinase family and in JAK1 across species (Supplementary Fig. 1). We studied localization of both mutations modelling JAK1 pseudokinase and kinase domains on the published TYK2 structure¹⁸. While P832S was located far from the kinase domain, P733L mapped in the β 7- β 8 loop close to the inter-domain interface and may affect interaction between the domains (Fig. 1d). Taken together, these results suggested that the newly found JAK1 genetic variants, especially P733L, may be recessive pathogenic mutations, rather than rare neutral polymorphisms.”

28. “Page 10, line 202: the mutant variant ?”

Corrected: “...the mutation found in the patient could either affect...”

29. “Suppl Fig. 2a and Fig 3: The level of JAK1 in patient's fibroblats seems definitely lower than in the two controls, as opposed to what the authors conclude. This is also seen in Fig. 3.”

Please see above our response to comment 15. We added new data in Suppl Fig 2.

30. “All JAK1 blots are overexposed. Quantification relative to actin should be presented in less exposed blots or in westerns with lower amount of lysate.”

We replaced all the over-exposed blots with less-exposed blots. We quantified bands either relative to actin or, for phosphorylated proteins, relative to their non-phosphorylated forms.

31. “Line 227 It cannot be concluded that the mutation P7333 “has an inhibitory effect” is not appropriate”

We rephrased:

“We found that phosphorylation of JAK1^{P832S} was similar to that of JAK1^{WT}, but phosphorylation of JAK1^{P733L/P832S} was completely abolished and that of JAK1^{P733L} was either abolished (Fig 7a) or very strongly reduced (Fig 7b), suggesting that the amino acid change P733L in the pseudokinase domain impairs JAK1 function leading to the reduced phosphorylation of tyrosines Y1034 and Y1035 in the activation loop of the kinase domain (Fig. 7a,b).”

32. “Suppl. Fig. 3: time of stimulation is not mentioned.”

We now included stimulation time in the figure legends (Suppl. Fig. 3 and Fig 4)

33. “Page 22, line 485: the second IFN γ in that line should read IFN α (Intron”

Corrected

Reviewer #2:

1. “The authors dismiss the possibility of any contribution to patient's condition from the other three genes identified with homozygous rare variants, which could potentially play a role. For example, one of these genes, ANXA1 (annexin A1) (Suppl Table 1), is involved in innate and adaptive immunity, as well as potential tumour-suppressor activity. Annexin A1 can also be linked to IFN- γ production, as this is increased in a knock out mouse model upon OVA stimulation (Yang, J. Immun., 2013).”

Although we cannot completely exclude modifier effects of other mutations, we note that mutation p.R235K in the ANXA1 gene is predicted by PolyPhen-2 to be Benign with a score 0 (on a scale 0 to 1), while both JAK1 mutations are predicted to be Probably Damaging with very high scores 0.997 and 0.999. We now included these predictions in Supplementary Table 1 and changed the text:

“Five of these rare variants were homozygous (Supplementary Table 1). Of these five, three were predicted to be benign, while two missense mutations were predicted to be probably damaging; both were located in the JAK1 gene, leading to amino-acid changes from proline to leucine (p.P733L) and from proline to serine (p.P832S) (Fig. 1b).”

2. “While it seems reasonable to associate JAK1 mutations to primary immunodeficiency (PID) via IFN- γ signalling, as there is literature linking mutations in IFN γ R1 and IFN γ R2 to PID, the link to cancer susceptibility is more tenuous. Unfortunately this manuscript lacks an in vivo model that would convincingly demonstrate that JAK1 P733L is driving the diseases seen in the patient, and this would be necessary to support the conclusions made by the

authors.”

We agree that having an *in vivo* model would be beneficial. However, we think that this is beyond the scope of the current manuscript. The importance of reporting a new primary immunodeficiency caused by the JAK1 mutations justifies publication of our current data. Given that patients with primary immunodeficiencies are rare, reports of single patients caused by new mutations advance the knowledge in the field, provided that functional effects of the putative mutations observed in isolated patients are clearly supported by the experimental data (e.g. see Casanova et al, Guidelines for genetic studies in single patients: lessons from primary immunodeficiencies, J Exp Med, 2014, 211(11): 2137-49).

We agree that our data support the causative role of the JAK1 mutations in primary immunodeficiency and susceptibility to mycobacterial infection observed in the patient. Also, we agree, that we cannot unambiguously link it with cancer (although we think it is important to emphasise the early age bladder in the patient’s clinical phenotype). Therefore, we rephrased the text, carefully stating that:
“Our findings clarify mechanisms of JAK1 signaling and demonstrate its critical role in protection against mycobacterial infection and, possibly, in immunological surveillance of cancer.”

3. “No attempt is made to discuss the effects of these amino acid changes at a structural level and how this might alter JAK1 function.”

We have now discussed potential effects of mutations on JAK1 structure and functions in Results:

“We studied localization of both mutations modelling JAK1 pseudokinase and kinase domains on the published TYK2 structure¹⁸. While P832S was located far from the kinase domain, P733L mapped in the β 7- β 8 loop close to the inter-domain interface and may affect interaction between the domains (Fig. 1d). ”

... and Discussion:

“In Janus kinases the pseudokinase domain stabilises the inactive conformation of the kinase domain⁴¹. In structural models of JAKs, many tumour-associated activating mutations map to the interface between pseudokinase and kinase domains, presumably disrupting interaction between them, which facilitates activation of the kinase domain^{18,41}. For example, JAK1 activating mutation F734L, found in a sample from a T-cell acute lymphoblastic leukemia patient^{42,43}, maps in the β 7- β 8 loop of the pseudokinase domain that interacts with the β 2- β 3 loop of the kinase domain^{18,41}. The P733L mutation that we found in our patient maps in the same pseudokinase β 7- β 8 loop next to F734L, however, as we have shown here, it reduces JAK1 activity, suggesting that P733L enhances auto-inhibitory interaction between the pseudokinase and the kinase domains.”

4. “It is not clear why the authors have chosen to show a homology-based model from 2008 (fig 1c), itself based on a prediction of JAK2, when crystal structures of JAK1 JH2 and JH1 are publically available. It would perhaps be more useful to map the mutations to the JH2 structure alone (PDB 4L00), or to model JAK1 JH1 and JH2 based the only crystal structure of a JAK to date with both kinase

and pseudokinase domains, that of TYK2 (PDB 4OLI). Although no full length crystal structures of JAK1 (or any other JAK) are currently available, there is some cryo-EM work on full length JAK1 (Lupardus, 2011) which is relevant to any discussion of JAK1 domain configuration.”

We now did both these analyses. We decided to include in Figure 1d JAK1 JH1 and JH2 domains modelled on the published structure of the TYK2 pseudokinase and kinase domains (PDB 4OLI). We also discuss positions of the P733L and P832S mutations (see response to the previous comment). In Discussion we also mentioned findings of the Lupardus et al 2011 paper:

“Although no full length crystal structure of JAK1 or any other Janus kinase is currently available, 2D electron microscopy averages and 3D reconstructions show that JAK1 domains have conformational flexibility and that pseudokinase and kinase domains are closely associated with each other⁴⁰”

Other points

5. “- In figure 2 it is not clear how many healthy controls were tested, or whether 1 healthy control was simply tested several times. It would be useful to have an idea of how much variation exists in a healthy population to be able to decide whether signalling defects exist in the patient's cells.”

In each case a sample of the patient was analysed against a healthy travel control that was taken simultaneously and delivered to the laboratory together with the patient's sample. In Fig 2a the analysis was done in triplicate: each dot represents an independent result. Some analyses were done once (3 dots shown), some (e.g. pSTAT1 after IFNa stimulation) were done on two different occasions (so there are 6 dots for the patient and 3+3 dots from two travel controls). In Fig 2b-2f the analysis was done similarly, but here we show data from a larger control population that was available to us in this case, because our clinical lab studied many controls over time, so we can show variation in a healthy population. We clarified this in the Figure legend.

6. “- pSTAT1 in U4A JAK1 P733L cells following IFN-g stimulation doesn't appear to be reduced compared to U4A JAK1 WT cells (Fig 5b), as claimed in the text (lines 230-231); if anything, it appears to be increased.”

We repeated this experiment several times. Indeed, reduction of STAT1 phosphorylation, particularly after IFNg stimulation was only clear for the double-mutant construct. The effect on STAT2 phosphorylation after IFNa stimulation was more obvious. This is now Fig 7 – please see the graphs. We also changed the text and removed this statement.

7. “- Molecular weight markers could be useful for the lower portion of fig. 6.”

We added molecular weight markers in this Figure (now Fig 8).

8. “- Also in Fig. 6, the results for the JAK1 kinase and pseudokinase domain deletion constructs are interesting, but is it possible to show a larger section of the Phospho-JAK1 blot that includes the region where you would expect to see JAK1ΨKinΔ and JAK1ΨKinΔ/K908E?”

Larger blot sections have been included in Fig 6 (now Fig 8). We couldn't detect any phosphorylated JAK1 in cells expressing those constructs (even in over-exposed films).

9. Additionally, the hierarchical transactivation model proposed here would be better supported with western blots using antibodies raised against the phosphorylated activation loops of JAK2 and TYK2.”

We attempted several times this analysis in U4C cells expressing different constructs using anti-phospho-JAK2 and anti-phospho-TYK2 antibodies, but we got very weak signals, so the results were not reliable and these experiments were not informative.

10. “- Even though the authors point out that over expression of JAKs are known to result in constitutive phosphorylation of the activation loop, it is not clear how using the U4C cell line would get around this, and indeed phosphorylation of JAK1 and STAT1 in the absence of cytokine are observed in U4C JAK1 WT, and to a lesser degree in U4C JAK1 P733L/P832S (Fig.6).”

Clones of U4C cells that we studied in this Figure (now Fig 8) stably expressed our JAK1 constructs inserted in the FRT sites. Such cells better respond to cytokine stimulation than transfected cells, e.g. on Fig 8 you can see that in cells expressing the wild type JAK1 levels of phospho-JAK1 increase after IFN γ or IFN α stimulation. This is in contrast to the stable levels of phospho-JAK1 in U4A cells that were transfected with wild-type JAK1 construct (Fig 7).

Reviewer #3 (Remarks to the Author):

“1. Given that the biochemical changes are expected to increase susceptibility to Mycobacteria the authors make only scant reference to the Mycobacterial infection of the patient. A better documentation is needed”

We have now extended description of the patient's disease with the available information (please see Results).

“2. While the representative gels in the Figures are quite clear the manuscript will be stronger if the results of the replicate experiments are shown in graphical quantitative plots next to the representative gels . This should be done at least for the major findings of decreased INF γ and INF α driven JAK1 and INF γ phosphorylation”

We agree and now have added graphs showing data quantification.

“3. The data on gene induction in Supplemental Fig 3 is important and should be shown as a standard Figure”

We now significantly expanded our analysis of gene expression induction using various doses of cytokines. The results are now shown in Figure 4 and Supplementary Fig. 3.

Reviewers' Comments:

Reviewer #1 (Remarks to the Author):

The revised manuscript is much improved, notably with the addition of qPCR data. Most of my comments have been addressed, except for the following ones:

2. “No measure of catalytic activity is provided.”

We measured kinase activity of JAK1 by studying STAT phosphorylation in primary fibroblasts and lymphocytes, as well as U4A and U4C cell lines. We now quantified protein phosphorylation in all experiments and present quantification results together with Western blot scans.

Referring to authors response

STAT1/2 phosphorylation in cells is not a measure of the catalytic activity of a JAK mutant. To be able to conclude that the P733L mutation affects JAK1 enzymatic activity by, for example, enhancing auto-inhibitory interaction between the pseudokinase and the kinase domains (line 379-381), the intrinsic catalytic activity should have been measured in an in vitro kinase assay (ie ability of immunopurified JAK1, wt and mutant, to auto-phosphorylate itself and phosphorylate a substrate when ATP is added).

3. “The major conclusions of the study is that the hypomorphic JAK1 is causative to the mycobacterial susceptibility and the development of the cancer in the patient. I do not think that these claims are substantiated by the analyses shown.”

We show that JAK1 mutations affect IFN signalling, leading to impaired STAT activation, including that of STAT1. It has been firmly established that mutations in this pathway, and specifically hypomorphic STAT1 mutations, lead to primary immunodeficiencies that manifest with susceptibility to mycobacterial infection (OMIM 614892 and 613796). Therefore, we believe that the causative link between JAK1 mutations and mycobacterial infection in our patient is quite clear. The link between the primary immunodeficiency caused by JAK1 mutations and cancer diagnosed in the patient is less certain. Thus, we carefully avoid suggesting a causative link between them (although we think that it is important to mention bladder carcinoma, as an unusual early feature in the patient):

Title: “Bi-allelic JAK1 mutations in immunodeficient patient with mycobacterial infection and cancer”

Abstract: “Our findings clarify mechanisms of JAK1 signaling and demonstrate its critical role in

protection against mycobacterial infection and, possibly, in immunological surveillance of cancer.”

Introduction: “...JAK1 mutation that leads to functional JAK1 deficiency associated with susceptibility to atypical mycobacterial infection and early-onset bladder carcinoma”

Also please see our response to Reviewer #2 comment 2 below.

Referring to authors response

Concerning the link between the JAK1 mutations and the early-onset bladder carcinoma : no experiments in this work address this possibility, no data support or exclude this association. Hence, it should not appear in the title.

5. “Experiments shown in Figure 6 do not provide any novel information on the role of JAK1 in IFNs signaling, nor on the defect of the mutants identified by the authors.”

In Figure 6 (now Fig 8) we studied stable clones of the JAK1-deficient U4C cells complemented with various JAK1 mutants. These cells did not show such a pronounced induction of the signalling cascade by mass activation as we observed in transiently transfected U4A (Fig 7).

Therefore, the stable clones provide a valuable addition to our experiments.

In Figure 8, there is no analysis of the single P733L mutant.

14. “Figure 2b-f

Whole blood was challenged for 24hr with either PHA or LPS and cytokine production (IFN γ , IL-10, TNF α , IL-6, IL-12) was measured by Elisa or multiplex. Are LPS responses increased ?”

After LPS stimulation production of IL-6 and TNF α was increased, while production of IFN γ and IL-12 was normal.

18. “At a very high dose of IFN α (100ng/ml), a defect in JAK1 phosphorylation is clearly observed in patient fibroblasts. Yet, the effect on STAT1 Pi is rather small.”

This observation is one of the findings of our study; we suggest that it is explained by the structural role of JAK1 in signal transduction that is independent of its phosphorylation:

“These observations suggest that JAK1 has a mode of function in interferon signalling that is independent of its phosphorylation. This putative JAK1 function would explain why such a profound defect in JAK1 phosphorylation observed in the patient-derived fibroblasts leads to only modest reduction in the downstream STAT phosphorylation (Fig. 3).”

It is not clear in what way the authors think that JAK1 contributes to STAT1 phosphorylation in a phosphorylation-independent manner ? What would be this structural role ?

23. “Figure 4:

here the authors conclude that TYK2 phosphorylation in response to IFN α is more affected than JAK2 phosphorylation in response to IFN γ . This comparison is not tenable. Different stimulating cytokines (for which level of cognate receptors likely differs) at non comparable doses were used. Affinity of phospho-specific antibodies can greatly vary.”

Of course, doses of IFN γ and IFN α used for stimulation, levels of receptors and affinity of antibodies differ between these cytokines. Therefore, we do not compare them directly. Instead, we study to what extent the responses to IFN α or IFN γ are affected in the patient relative to control cells stimulated with the same cytokine. In Fig 4 (now Fig 6) we can clearly see that in response to IFN α stimulation the level of phospho-TYK2 is strongly reduced in the patient relative to control, while the level of phospho-JAK2 is only slightly reduced in the patient relative to control after IFN γ stimulation (please see graphs in Fig 6) .

Fig. 6 of revised version : I still think that the conclusion that that « TYK2 phosphorylation is strongly reduced while JAK2 phosphorylation is preserved » is not appropriate. Quantification of the phosphorylation of the juxtaposed kinase should take into consideration the level of JAK1 wt and mutant phosphorylation in the two experimental settings.

These data are now provided (level of JAK1 phosphorylation in panels a and b) and show a reduction in JAK1 P733L phosphorylation which appears to proportionally affect the level of TYK2 and JAK2 phosphorylations.

Activation of JAK1 P733L is impaired and the phosphorylation of the juxtaposed kinase appears proportionally impaired.

26. “Quite convincingly JAK1 P7333L is not phosphorylated, neither basally nor after cytokine stimulation and yet STAT1 phosphorylation in these transfected cells is very weakly affected. Stable clones should be analyzed.”

We analysed stable clones; the results are shown in Figure 8.

In Figure 8, there is no analysis of the single P733L mutant.

Reviewer #2 (Remarks to the Author):

The manuscript is substantially improved and makes a very compelling case, both for the immunodeficiency part and for a different role of JAK1 in the type I and type II IFN complexes. The work has relevance for the basic field of JAK function and signaling and I would accept it. The only minor issue is that when authors say that there is a phosphorylation-independent mode for JAK1 function in IFN gamma via JAK2, they need to specify whether they refer to the phosphorylation of activation loop tyrosines only, or globally to all pY sites in JAK1, or to both.

Reviewer #3 (Remarks to the Author):

I am satisfied that the authors addressed my concerns. No further comments

Reviewers' comments are shown in bold. Our new responses are indicated by:

→

Reviewer #1 (Remarks to the Author):

The revised manuscript is much improved, notably with the addition of qPCR data. Most of my comments have been addressed, except for the following ones:

2. “No measure of catalytic activity is provided.”

We measured kinase activity of JAK1 by studying STAT phosphorylation in primary fibroblasts and lymphocytes, as well as U4A and U4C cell lines. We now quantified protein phosphorylation in all experiments and present quantification results together with Western blot scans.

Referring to authors response

STAT1/2 phosphorylation in cells is not a measure of the catalytic activity of a JAK mutant. To be able to conclude that the P733L mutation affects JAK1 enzymatic activity by, for example, enhancing auto-inhibitory interaction between the pseudokinase and the kinase domains (line 379-381), the intrinsic catalytic activity should have been measured in an in vitro kinase assay (ie ability of immunopurified JAK1, wt and mutant, to auto-phosphorylate itself and phosphorylate a substrate when ATP is added).

→ New response:

In such an in vitro assay the catalytic activity of the immunoprecipitated JAK1 is analyzed in cell-free conditions. The experiments that we present in the manuscript studied the function of the mutant JAK1 in its natural intracellular environment, i.e. in interaction with partner proteins that regulate its activity. Although our experiments did not directly measure the catalytic activity of JAK1, they assessed its biologically relevant effects (e.g. levels of phosphorylated STAT proteins) and demonstrated functional deficiency of the mutant JAK1 protein.

To make it clear that we refer to the function of the mutant JAK1 rather than specifically to its kinase activity, we rephrased the text:

“The P733L mutation that we found in our patient maps in the same pseudokinase β 7- β 8 loop next to F734L, however, as we have shown here, P733L reduces JAK1-mediated signaling, suggesting that it may enhance auto-inhibitory interaction between the pseudokinase and the kinase domains.” (page 17)

3. “The major conclusions of the study is that the hypomorphic JAK1 is causative to the mycobacterial susceptibility and the development of the cancer in the patient. I do not think that these claims are substantiated by the analyses shown.”

We show that JAK1 mutations affect IFN signalling, leading to impaired STAT

activation, including that of STAT1. It has been firmly established that mutations in this pathway, and specifically hypomorphic STAT1 mutations, lead to primary immunodeficiencies that manifest with susceptibility to mycobacterial infection (OMIM 614892 and 613796). Therefore, we believe that the causative link between JAK1 mutations and mycobacterial infection in our patient is quite clear. The link between the primary immunodeficiency caused by JAK1 mutations and cancer diagnosed in the patient is less certain. Thus, we carefully avoid suggesting a causative link between them (although we think that it is important to mention bladder carcinoma, as an unusual early feature in the patient):

Title: “Bi-allelic JAK1 mutations in immunodeficient patient with mycobacterial infection and cancer”

Abstract: “Our findings clarify mechanisms of JAK1 signaling and demonstrate its critical role in protection against mycobacterial infection and, possibly, in immunological surveillance of cancer.”

Introduction: “...JAK1 mutation that leads to functional JAK1 deficiency associated with susceptibility to atypical mycobacterial infection and early-onset bladder carcinoma”

Also please see our response to Reviewer #2 comment 2 below.

Referring to authors response

Concerning the link between the JAK1 mutations and the early-onset bladder carcinoma : no experiments in this work address this possibility, no data support or exclude this association. Hence, it should not appear in the title.

→ New response:

Yes, this is a fair point. We revised the title to: “Bi-allelic JAK1 mutations in immunodeficient patient with mycobacterial infection”

5. “Experiments shown in Figure 6 do not provide any novel information on the role of JAK1 in IFNs signaling, nor on the defect of the mutants identified by the authors.”

In Figure 6 (now Fig 8) we studied stable clones of the JAK1-deficient U4C cells complemented with various JAK1 mutants. These cells did not show such a pronounced induction of the signalling cascade by mass activation as we observed in transiently transfected U4A (Fig 7). Therefore, the stable clones provide a valuable addition to our experiments.

In Figure 8, there is no analysis of the single P733L mutant.

→ New response:

In Fig 8 we studied the double mutant form JAK1^{P733L/P832S} that was actually found in the patient and confirmed that it is functionally deficient in the U4C cell model (similarly to primary fibroblasts and U4A cells studied in previous figures). We did not include here single-mutant forms JAK1^{P733L} and JAK1^{P832S} for these reasons:

- the experiment shown in Fig 8 is quite big already, so we didn't want overcomplicate it;

- in Fig 7 we already compared JAK1^{P733L} and JAK1^{P832S} with the patient's double-mutant JAK1^{P733L/P832S} and the wild type JAK1 using various doses of cytokines, which is more comprehensive than stimulations with a single dose used in Fig 8;

- the main message of Fig 8 is different - it shows that in the presence of its pseudo-kinase domain JAK1 without the kinase domain and the kinase-dead JAK1 can still mediate phosphorylation of STAT1.

Therefore, we think that adding analysis of single-mutant JAK1^{P733L} and JAK1^{P832S} in Fig 8 wouldn't be helpful, as it is (a) somewhat redundant, and (b) will divert attention from the main purpose of this figure.

Overall, we feel that in our study it was more appropriate to focus on the effects of the double-mutant JAK1^{P733L/P832S}, because that is the protein that actually caused immunodeficiency. Hence, differential analysis of single mutants is limited to one section and Fig 7.

14. "Figure 2b-f

Whole blood was challenged for 24hr with either PHA or LPS and cytokine production (IFN γ , IL-10, TNF α , IL-6, IL-12) was measured by Elisa or multiplex. Are LPS responses increased ?"

After LPS stimulation production of IL-6 and TNF α was increased, while production of IFN γ and IL-12 was normal.

→ New response:

The response to LPS stimulation in the patient's blood was abnormal: it was increased in terms of IL-6 and TNF α secretion, although production of other cytokines, IFN γ and IL-12, was not significantly increased, as shown in Fig 2b-f.

18. "At a very high dose of IFN α (100ng/ml), a defect in JAK1 phosphorylation is clearly observed in patient fibroblasts. Yet, the effect on STAT1 Pi is rather small."

This observation is one of the findings of our study; we suggest that it is explained by the structural role of JAK1 in signal transduction that is independent of its phosphorylation:

"These observations suggest that JAK1 has a mode of function in interferon signalling that is independent of its phosphorylation. This putative JAK1 function would explain why such a profound defect in JAK1 phosphorylation observed in the patient-derived fibroblasts leads to only modest reduction in the downstream STAT phosphorylation (Fig. 3)."

It is not clear in what way the authors think that JAK1 contributes to STAT1 phosphorylation in a phosphorylation-independent manner ? What would be this structural role ?

→ New response:

Our data show that JAK1 plays a structural role, which is independent of its phosphorylation and kinase function. For example, it may provide a scaffold to the partner Janus kinase, although the exact nature of this interaction is not clear.

To clarify, we now added in the text:

“These observations suggest that JAK1 has a mode of function in interferon signaling that is independent of phosphorylation of tyrosines in its activation loop, e.g. providing a scaffold for the juxtaposed Janus kinase.” (page 13)

23. “Figure 4:

here the authors conclude that TYK2 phosphorylation in response to IFN α is more affected than JAK2 phosphorylation in response to IFN γ . This comparison is not tenable. Different stimulating cytokines (for which level of cognate receptors likely differs) at non comparable doses were used. Affinity of phospho-specific antibodies can greatly vary.”

Of course, doses of IFN γ and IFN α used for stimulation, levels of receptors and affinity of antibodies differ between these cytokines. Therefore, we do not compare them directly. Instead, we study to what extent the responses to IFN α or IFN γ are affected in the patient relative to control cells stimulated with the same cytokine. In Fig 4 (now Fig 6) we can clearly see that in response to IFN α stimulation the level of phospho-TYK2 is strongly reduced in the patient relative to control, while the level of phospho-JAK2 is only slightly reduced in the patient relative to control after IFN γ stimulation (please see graphs in Fig 6) .

Fig. 6 of revised version : I still think that the conclusion that that « TYK2 phosphorylation is strongly reduced while JAK2 phosphorylation is preserved » is not appropriate. Quantification of the phosphorylation of the juxtaposed kinase should take into consideration the level of JAK1 wt and mutant phosphorylation in the two experimental settings.

These data are now provided (level of JAK1 phosphorylation in panels a and b) and show a reduction in JAK1 P733L phosphorylation which appears to proportionally affect the level of TYK2 and JAK2 phosphorylations. Activation of JAK1 P733L is impaired and the phosphorylation of the juxtaposed kinase appears proportionally impaired.

→ New response:

Yes, we agree that levels of phospho-TYK2 and phospho-JAK2 correlate with levels of phospho-JAK1. We have now added phospho-JAK1 graphs in Fig 6 to illustrate this. We also rephrased the text. Basically, here we just describe the data shown in Fig. 6:

“We found that in patient’s fibroblasts the amount of phosphorylated TYK2 was strongly reduced, while the amount of phosphorylated JAK2 was only slightly diminished (Fig. 6).” (page 12)

We discuss possible reasons for this difference in Discussion (page 17).

26. “Quite convincingly JAK1 P7333L is not phosphorylated, neither basally nor after cytokine stimulation and yet STAT1 phosphorylation in these transfected cells is very weakly affected. Stable clones should be analyzed.”

We analysed stable clones; the results are shown in Figure 8.

In Figure 8, there is no analysis of the single P733L mutant.

→ New response:

Please see our response to comment 5 above.

Reviewer #2 (Remarks to the Author):

The manuscript is substantially improved and makes a very compelling case, both for the immunodeficiency part and for a different role of JAK1 in the type I and type II IFN complexes. the work has relevance for the basic field of JAK function and signaling and I would accept it. The only minor issue is that when authors say that there is a phosphorylation-independent mode for JAK1 function in IFN gamma via JAK2, they need to specify whether they refer to the phosphorylation of activation loop tyrosines only, or globally to all pY sites in JAK1, or to both.

→ New response:

As we measured phosphorylation of tyrosines in the activation loop, we clarified:
“These observations suggest that JAK1 has a mode of function in interferon signaling that is independent of phosphorylation of tyrosines in its activation loop.” (page 13)

Reviewer #3 (Remarks to the Author):

I am satisfied that the authors addressed my concerns. No further comments

Other changes

To comply with the manuscript format we also edited subheadings in Results. All changes are shown in yellow.

Reviewer #1 (Remarks to the Author):

The authors have answered the points raised.